# An axonemal intron splicing program sustains *Plasmodium* male development

Jiepeng Guan[1,4], Peijia Wu[1,4], Xiaoli Mo[1,4], Xiaolong Zhang[2,4], Wenqi Liang[1], Xiaoming Zhang[1], Lubin Jiang[2] ✉, Jian Li [1] ✉, Huiting Cui[1] ✉ & Jing Yuan [1,3] ✉

Differentiation of male gametocytes into flagellated fertile male gametes relies on the assembly of axoneme, a major component of male development for mosquito transmission of the malaria parasite. RNA-binding protein (RBP)-mediated post-transcriptional regulation of mRNA plays important roles in eukaryotic sexual development, including the development of female *Plasmodium*. However, the role of RBP in defining the *Plasmodium* male transcriptome and its function in male gametogenesis remains incompletely understood. Here, we performed genome-wide screening for gender-specific RBPs and identified an undescribed male-specific RBP gene *Rbpm1* in the *Plasmodium*. RBPm1 is localized in the nucleus of male gametocytes. RBPm1-deficient parasites fail to assemble the axoneme for male gametogenesis and thus mosquito transmission. RBPm1 interacts with the spliceosome E complex and regulates the splicing initiation of certain introns in a group of 26 axonemal genes. RBPm1 deficiency results in intron retention and protein loss of these axonemal genes. Intron deletion restores axonemal protein expression and partially rectifies axonemal defects in RBPm1-null gametocytes. Further splicing assays in both reporter and endogenous genes exhibit stringent recognition of the axonemal introns by RBPm1. The splicing activator RBPm1 and its target introns constitute an axonemal intron splicing program in the post-transcriptional regulation essential for *Plasmodium* male development.

Malaria is a worldwide infectious disease caused by the protozoan parasite *Plasmodium*[1]. The spread of *Plasmodium* depends on the transition between the mammal host and the *Anopheles* mosquito. In mammal hosts, a small proportion of intraerythrocytic asexual parasites undergo sexual development, irreversibly differentiating into the sexual precursor gametocytes, which are transmission-competent for the mosquito vector[2]. Within 10 min after being ingested into the mosquito midgut, the gametocytes escape from host erythrocytes and develop into fertile gametes, a process known as gametogenesis[3]. A flagellated motile male gamete fertilizes with a female gamete to form a zygote. After the zygote-ookinete-oocyst-sporozoite development in

mosquitoes, the parasites are finally injected from the salivary gland into a mammal host, completing the transmission of the malaria parasite[4].

Sexual development plays a central role in malaria transmission[5,6]. When activated by two joint environmental stimuli (a temperature drop[7] and a metabolite xanthurenic acid[8]) in the mosquito midgut, a female gametocyte produces a haploid gamete, while a male gametocyte gives rise to 8 haploid gametes[3]. Female gametogenesis undergoes minor morphological changes, while male gametogenesis involves fast and spectacular changes[9,10]. During the male gametogenesis, two spatially distinct components are coordinated. One is the

[1]State Key Laboratory of Cellular Stress Biology, School of Life Sciences, Faculty of Medicine and Life Sciences, Xiamen University, Xiamen, China. [2]Shanghai Institute of Immunity and Infection, Chinese Academy of Sciences, Shanghai, China. [3]Department of Infectious Disease, Xiang'an Hospital of Xiamen University, School of Medicine, Faculty of Medicine and Life Sciences, Xiamen University, Xiamen, China. [4]These authors contributed equally: Jiepeng Guan, Peijia Wu, Xiaoli Mo, Xiaolong Zhang. ✉e-mail: lbjiang@siii.cas.cn; jianli_204@xmu.edu.cn; cuihuiting@xmu.edu.cn; yuanjing@xmu.edu.cn

cytoplasmic assembly of 8 basal bodies and axonemes, and the other is the 3 successive rounds of genome replication without nuclear division, resulting in an octoploid nucleus. Subsequently, 8 axonemes with chromosomes attached are released from the cell body of male gametocytes, resulting in 8 flagellated daughter gametes, which is the process termed "exflagellation".

Parasite stage transition during the *Plasmodium* life cycle requires a fine-tuned multilayer regulation of gene expression[11–13]. Previous studies have identified transcriptional and epigenetic programs critical for the sexual commitment and development of the gametocytes[14–23]. However, how the *Plasmodium* establishes distinct repertoires of transcripts between male and female gametocytes remains incompletely illustrated. RNA-binding proteins (RBPs) can interact with transcripts in all manner of RNA-driven processes[24]. RBPs regulate all aspects of the life cycle of mRNA, including mRNA transcription, splicing, modification, trafficking, translation, and decay[25,26]. RBP-containing ribonucleoprotein complexes, such as the DOZI (development of zygote inhibited) complex and CITH (CARI/Trailer Hitch homolog) complex, had been shown to repress the translation of multiple mRNAs in female gametocytes[27–29]. So far, our understanding of post-transcription control is still limited in the male gametogenesis. Recent transcriptome studies in both human malaria parasite *P. falciparum* and mouse malaria parasite *P. berghei* revealed that certain RBPs are specifically or preferentially expressed in male gametocytes[30,31], implying gender-specific roles of RBPs in the post-transcriptional regulation for male development. However, systematic identification of male RBPs for male gametogenesis and their precise roles in defining the gender distinct transcriptome via the post-transcription regulation have not been reported.

In this work, we perform comparative transcriptome analysis on male and female gametocytes and obtain a list of gender-specific RBPs in the rodent malaria parasite *P. yoelii*. From this list, we identify a functionally unknown gene (PY17X_0716700, named as *Rbpm1* in this study), which is specifically transcribed in male gametocytes. We demonstrate that RBPm1 is a nuclear RBP essential for male gametogenesis and mosquito transmission of parasite. RBPm1 interacts with the spliceosome E complex and initiates the splicing of certain introns in a group of axonemal genes. RBPm1-deficient parasites cannot express these axonemal proteins and fail to assemble the axoneme. These findings reveal an RBPm1-mediated intron splicing program of the axonemal genes essential for *Plasmodium* male development.

## Results

### RNA-binding protein RBPm1 is expressed in the nucleus of male gametocytes

Approximately 180 putative *Plasmodium* RBPs had been predicted in silico[32]. To identify the key RBPs for male gametogenesis, we searched the male gametocyte-specific RBPs in the rodent malaria parasite *P. yoelii*. Using the fluorescence-activated cell sorting, highly purified male and female gametocytes were collected from a *P. yoelii* reporter line *DFsc7* (Fig. 1A and Supplementary Fig. 1A), in which fluorescent proteins GFP and mCherry are expressed in male and female gametocytes, respectively[33]. We performed RNA-seq and obtained gender-specific gametocyte transcriptome (Supplementary Fig. 1B and Supplementary Data 1). Among the 179 *P. yoelii* RBPs, an unstudied gene (PY17X_0716700) was identified with the greatest enrichment in male compared to female gametocytes (Fig. 1B, left panel). This gene was named as *Rbpm1* for RBP in male gametocyte. Notably, the *Rbpm1* orthologs PBANKA_0716500 and PF3D7_0414500 are also among the top male RBP genes of *P. berghei* and *P. falciparum*, respectively, (Fig. 1B, middle and right panels) based on the gender gametocyte transcriptomes[30,31].

To investigate RBPm1 expression during the parasite life cycle, we tagged endogenous RBPm1 with a sextuple HA (6HA) at the carboxyl (C)-terminus in the *P. yoelii* 17XNL strain (wild type or WT) using CRISPR-

Cas9[34,35]. The tagged parasite *Rbpm1::6HA* developed normally in mice and mosquitoes, indicating no detectable detrimental effect of tagging on protein function. Immunofluorescent assay (IFA) showed that RBPm1 was expressed only in gametocytes, but not in asexual blood stages, ookinetes, oocysts, or sporozoites (Fig. 1C, upper panel). Immunoblot also confirmed the gametocyte-restricted expression of RBPm1 (Fig. 1D). Gametocyte-specific expression of RBPm1 was observed in another parasite line *Rbpm1::gfp*, in which RBPm1 was tagged with GFP from the 17XNL (Fig. 1C, lower panel). To dissect whether RBPm1 expression is male-specific, the *Rbpm1::6HA* gametocytes were co-stained with antibodies against HA and α-Tubulin II (a highly expressed protein in male gametocytes[36]. RBPm1 was only detectable in the male gametocytes (Fig. 1E). Additionally, we tagged RBPm1 with 6HA in the reporter line *DFsc7* and observed the male-specific expression of RBPm1 (Fig. 1F). We noticed the nuclear localization of RBPm1 in all the male gametocytes tested (Fig. 1C, E, F), which was further confirmed by immunoblot of nuclear and cytoplasmic fractions from the *Rbpm1::6HA* gametocytes (Fig. 1G). Last, we analyzed the localization dynamics of RBPm1 throughout the process of gametogenesis (0, 2, 8, and 15 min post activation, mpa) in the *Rbpm1::6HA* parasites. Both IFA and immunoblot revealed consistent protein expression profile and nuclear localization of RBPm1 during gametogenesis (Fig. 1H, I). Together, these results demonstrated that RBPm1 was a nuclear protein specifically expressed in the male gametocytes.

### RBPm1 is essential for male gametogenesis and mosquito transmission of parasite

*P. yoelii Rbpm1* gene encodes a protein of 361 amino acid (aa) residues, with two RNA recognition motifs (RRM1 and RRM2). To investigate its function, we generated a mutant line, *ΔRbpm1*, by deleting the entire genomic sequence (1904 bp) of *Rbpm1* gene in *P. yoelii* 17XNL strain using CRISPR-Cas9 (Fig. 2A). *ΔRbpm1* produced normal level of male and female gametocytes in mice (Fig. 2B), indicating that RBPm1 is not essential for asexual blood stage proliferation and gametocyte formation. We next measured the male gametogenesis by counting exflagellation centers (ECs) in vitro after stimulation with 50 μM xanthurenic acid (XA) at 22 °C. *ΔRbpm1* showed a striking deficiency in the EC formation (Fig. 2C, D) and male gamete release (Fig. 2E). In contrast, RBPm1 disruption had no impact on female gamete formation in vitro (Fig. 2F), which corresponded with no RBPm1 expression in female. *ΔRbpm1* produced no ookinetes in vitro (Fig. 2G) or midgut oocysts and salivary gland sporozoites in the infected mosquitoes (Fig. 2H, I), indicating transmission failure in mosquito. Additionally, we deleted each of the RNA recognition motifs RRM1 (119–190 aa) and RRM2 (203-274 aa) of endogenous RBPm1 in the 17XNL (Fig. 2A). Both mutants, *Δrrm1* and *Δrrm2*, displayed similar defects as those observed in *ΔRbpm1* (Fig. 2B, D), suggesting essential role of both two RNA recognition motifs in RBPm1 function.

To further confirm that the *ΔRbpm1* phenotype was caused by *Rbpm1* deficiency, we introduced a sequence consisting of the coding region of *Rbpm1* and a N-terminal quadruple Myc epitope (4Myc) back to the *Rbpm1* locus in the *ΔRbpm1* line, generating the complemented line, referred to as *rescue* (Fig. 2A). The 4Myc-tagged RBPm1 was detected in the *rescue* gametocytes (Fig. 2J) and localized in the nucleus of male gametocytes (Fig. 2K). The *rescue* parasites restored the formation of ECs (Fig. 2C, D), male gametes (Fig. 2E), ookinetes (Fig. 2G), midgut oocysts (Fig. 2H), and salivary gland sporozoites (Fig. 2I).

Lastly, we performed genetic crosses between *ΔRbpm1* mutant and the male-deficient line *Δmap2* or the female-deficient line *Δnek4*. As expected, the cross between *Δmap2* and *Δnek4* produced the ookinetes in vitro (Fig. 2L). The ookinete formation was restored in the *ΔRbpm1* parasites that were crossed with *Δnek4* but not *Δmap2*, further confirming the defective male gamete formation in the *ΔRbpm1*. Together, these results demonstrated that RBPm1 is essential for male gametogenesis and mosquito transmission of parasites.

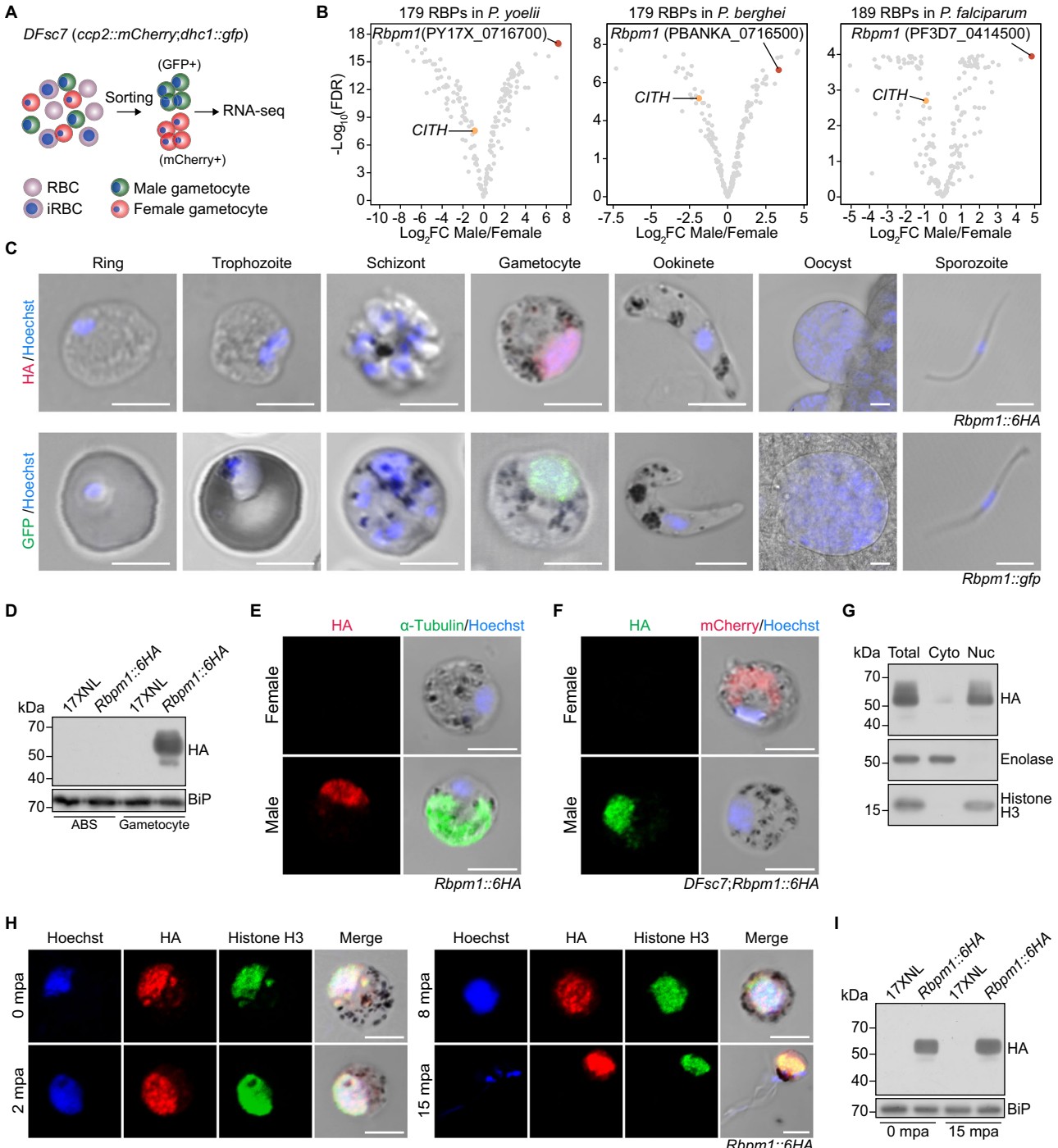

**Fig. 1 | RNA-binding protein RBPm1 is expressed in the nucleus of male gametocytes. A** Flowchart showing the purification and transcriptome analysis of male (green, GFP+) and female (red, mCherry+) gametocytes from a *P. yoelii* parasite reporter line *DFsc7*. **B** Gender analysis of gene transcription for the *Plasmodium* genome-wide putative RBPs between male and female gametocytes. The top male gene PY17X_0716700, *RBPm1*, is marked in red. CITH (orange dot) is a known female RBP. The results of *P. berghei* (middle panel) and *P. falciparum* (right panel) were based on the published gametocyte transcriptomes contributed by Yeoh, L.M. 2017 and Lasonder, E. 2016. The *p*-values were calculated by quasi-likelihood F-test and adjusted by false discovery rate (FDR). **C** Stage expression of RBPm1 during the *P. yoelii* life cycle. Immunofluorescence assay (IFA) of RBPm1 expression in the *Rbpm1::6HA* parasites stained with anti-HA antibody (top panel). Live cell imaging of the RBPm1::GFP protein in the *Rbpm1::gfp* parasites (bottom panel). Nuclei were stained with Hoechst 33342. Three independent experiments with similar results. Scale bars: 5 µm. **D** Immunoblot of RBPm1 in the asexual blood stage (ABS) and gametocyte of the *Rbpm1::6HA* parasites. BiP as a loading control. Three independent experiments with similar results. **E** IFA of HA-tagged RBPm1 and α-Tubulin (male gametocyte marker protein) in *Rbpm1::6HA* gametocytes. Three independent experiments with similar results. Scale bars: 5 µm. **F** IFA of HA-tagged RBPm1 and mCherry (expressed in female gametocytes) in the *DFsc7;Rbpm1::6HA* gametocytes. Three independent experiments with similar results. Scale bars: 5 µm. **G** Immunoblot of RBPm1 in cytosolic and nuclear fractions of *Rbpm1::6HA* gametocytes. Enolase (cytoplasmic/Cyto) and histone H3 (nuclear/Nuc) proteins used as controls respectively. Two independent experiments with similar results. **H** IFA of HA-tagged RBPm1 and histone H3 during male gametogenesis of the *Rbpm1::6HA* parasites. mpa: minute post activation. Three independent experiments with similar results. Scale bars: 5 µm. **I** Immunoblot of RBPm1 expression in the *Rbpm1::6HA* parasites during male gametogenesis. Two independent experiments with similar results.

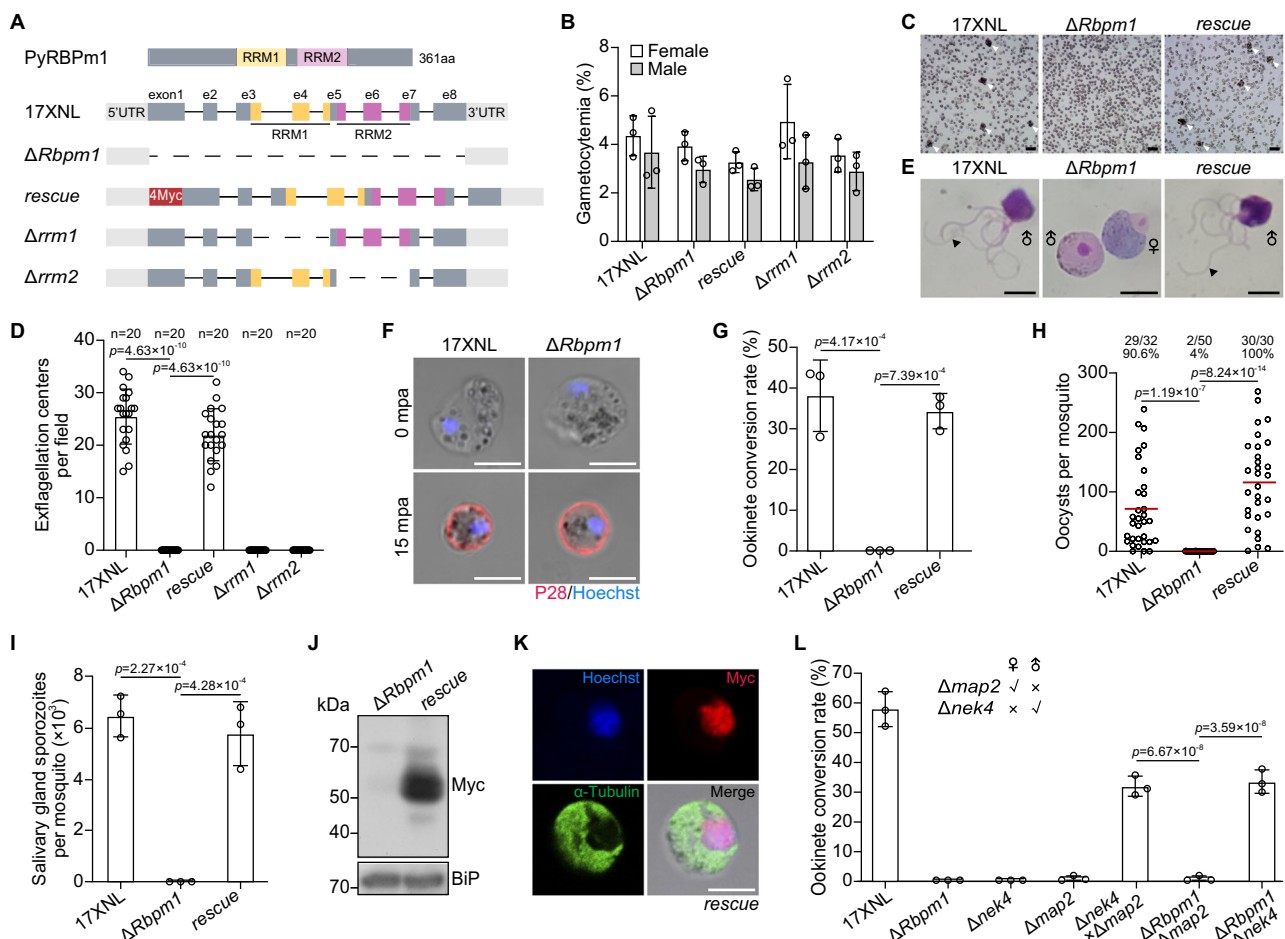

**Fig. 2 | RBPm1 is essential for male gametogenesis and mosquito transmission of parasite. A** A schematic showing genetic modification at the *Rbpm1* locus in the *P. yoelii* parasite. The top panel depicts the protein structure of RBPm1 with two RNA recognition motifs RRM1 (residues 119-190, yellow) and RRM2 (residues 203-274, purple). Δ*Rbpm1*, deletion of the whole coding sequence from the 17XNL (wild type) strain; *rescue*, the Δ*Rbpm1* line complemented with *Rbpm1* fused with a 4Myc tag; Δ*rrm1* and Δ*rrm2*, deletion each of RRM1 and RRM2 from the 17XNL. **B** Female and male gametocyte formation in mice for the modified parasite. Data are means ± SEM of three independent experiments. **C** Exflagellation center (EC) formation of activated male gametocytes at 10 mpa. Cell clusters representing the EC are marked with white arrows. Four independent experiments with similar results. Scale bars: 20 μm. **D** Quantification of EC formation. The ECs were counted within a 1 × 1-mm square area in the hemocytometer under a light microscope. n represents the number of fields counted. Means ± SEM, one-way ANOVA with Tukey multiple pairwise-comparisons. Three independent experiments. **E** Light microscope images of the exflagellated male gametes (black arrow) after Giemsa staining. Four independent experiments with similar results. Scale bars: 5 μm. **F** Female gamete formation assayed by P28 staining. P28 is a female gamete plasma membrane protein. *n* = 32 and 35 female gametocytes in 17XNL and Δ*Rbpm1* respectively. Scale bars: 5 μm. **G** Ookinete formation in vitro. Data are means ± SEM from three

independent experiments, one-way ANOVA with Tukey multiple pairwise-comparisons. **H** Midgut oocyst formation in mosquitoes at 7 days after blood feeding. x/y at the top represents the number of mosquitoes containing oocysts/ the number of dissected mosquitoes, and the percentage represents the infection prevalence of mosquitoes. Red lines show the mean value of oocyst numbers, one-way ANOVA with Tukey multiple pairwise-comparisons. Three independent experiments with similar results. **I** Salivary gland sporozoite formation in mosquitoes at 14 days after blood feeding. At least 20 infected mosquitoes were counted in each group. Data are means ± SEM of three independent experiments, one-way ANOVA with Tukey multiple pairwise-comparisons. **J** Immunoblot analysis of RBPm1 expression in gametocytes of the complemented line *rescue*. BiP as a loading control. Two independent experiments with similar results. **K** IFA of Myc-tagged RBPm1 and α-Tubulin in gametocytes of the *rescue* parasite. Three independent experiments with similar results. Scale bars: 5 μm. **L** Gender gamete fertility assay of the Δ*Rbpm1* by parasite genetic cross. Fertility was determined by ookinete development of Δ*Rbpm1* gametes after cross-fertilization with mutant lines that are defective in either female (Δ*nek4*) or male (Δ*map2*) gametes. Data are means ± SEM of three independent experiments, one-way ANOVA with Tukey multiple pairwise-comparisons.

## Defective axoneme assembly in RBPm1-deficient male gametogenesis

Next, we delineated more detailed defects of Δ*Rbpm1*. During male gametogenesis, the parasites undergo axoneme assembly, genome replication, rupture of the parasitophorous vacuole membrane (PVM) and erythrocyte membrane (EM), and finally releasing eight uni-flagellated male gametes. We first assessed the axoneme assembly. At 0 mpa, both α- and β-Tubulin were evenly distributed in the cytosol of male gametocytes of WT and Δ*Rbpm1* (Fig. 3A, upper panel). Immunoblot also detected comparable level for both Tubulins in gametocytes between WT and Δ*Rbpm1* (Fig. 3B). At 8 mpa, the axonemal

microtubules (MTs) were observed to be coiled around the enlarged nucleus in the WT gametocytes. However, aberrant axonemes were formed in Δ*Rbpm1* (Fig. 3A, middle panel). At 15 mpa, Δ*Rbpm1* failed to produce flagellated male gametes (Fig. 3A, lower panel). Under ultra-structure expansion microscopy (U-ExM)[37], the axonemes lost bundled structures at 8 mpa in Δ*Rbpm1* compared to the organized axonemes in WT (Fig. 3C). We used electron microscope to dissect the ultra-structural defects of axoneme in Δ*Rbpm1* male gametocytes at 8 mpa. The majority of axonemes (93%, 150 axonemes from 43 section images) displayed 9 + 2 arrangement of MTs in WT (Figs. 3D). In contrast, no intact axonemes (from 69 section images) were detected in either

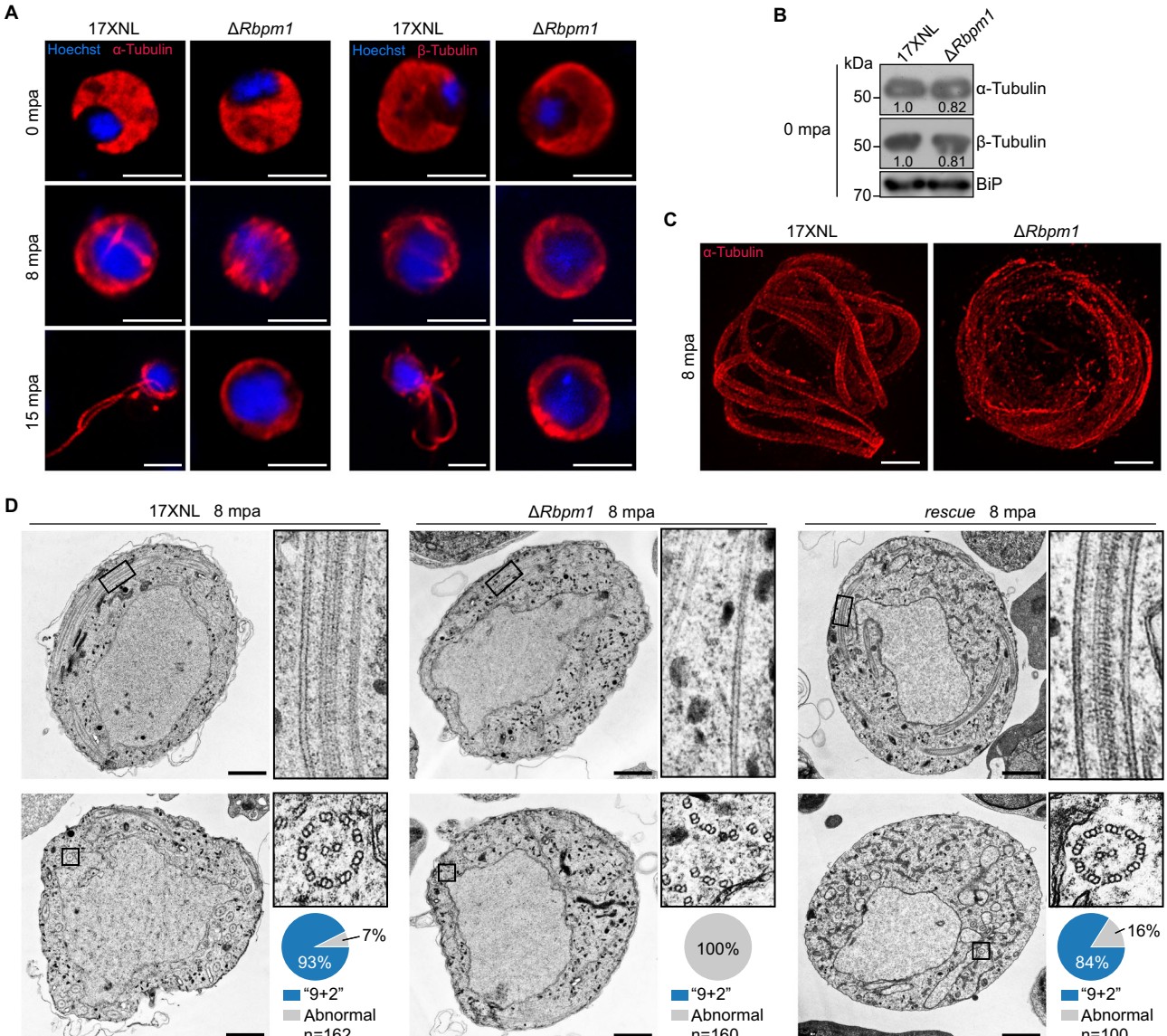

**Fig. 3 | Defective axonemal assembly in RBPm1-null male gametogenesis.**
**A** Detection of formation and exflagellation of axonemes during male gameto-genesis (0, 8, and 15 mpa) by staining α-Tubulin (left panels) and β-Tubulin (right panels). Nuclei were stained with Hoechst 33342. Four independent experiments with similar results. Scale bars: 5 μm. **B** Immunoblot of α- and β-Tubulins in gametocytes. The numbers indicate the relative intensities of the bands in the immunoblots. BiP as a loading control. Two independent experiments with similar results. **C** Ultrastructure expansion microscopy (U-ExM) of the axonemes in male gametocytes stained with α-Tubulin antibody at 8 mpa. Three independent experiments with similar results. Scale bars: 5 μm. **D** Transmission electron microscopy of axoneme architecture in male gametocytes at 8 mpa. Inset panels show longitudinal sections (top panels) and cross sections (bottom panels) of axonemes. The enclosed area (black box) was zoomed in. Pie charts show the quantification of axoneme ("9 + 2" microtubules) in the mutant parasites. n is the total number of intact and defective axoneme structures observed in each group. Three independent experiments with similar results. Scale bars: 1 μm.

longitudinal or cross sections of Δ*Rbpm1* (Figs. 3D). All the axonemes in Δ*Rbpm1* showed severe defects with loss of either central singlet MTs or peripheral doublet MTs (Fig. 3D), consistent with the obser-vation of Tubulin staining in Fig. 3C. In the complemented line *rescue*, the axoneme assembly restored to normal as in WT (Figs. 3D). These results demonstrated that RBPm1 is required for axoneme assembly during male gametogenesis.

We additionally analyzed genome replication and erythrocyte rupture during male gametogenesis. Flow cytometry analysis of male gametocytes at 8 mpa detected a comparable increase in DNA content in both parental *DFsc7* and its derivative mutant *DFsc7;*Δ*Rbpm1* para-sites (Supplementary Fig. 2A). These results indicated normal genome replication in the absence of RBPm1, consistent with the enlarged nucleus observed in the activated Δ*Rbpm1* male gametocytes from both the fluorescence and electron microscope images (see Fig. 3A, D).

In addition, immunostaining of SEP1 (parasite PVM protein) and TER119 (mouse EM protein) showed that RBPm1 deficiency had no notable effect on parasite rupture from the gametocyte-residing ery-throcytes (Supplementary Fig. 2B, C).

## RBPm1 deficiency causes defective intron splicing of axonemal genes
To investigate the mechanism of RBPm1 in regulating the axoneme assembly, we performed RNA-seq to examine the changes in male transcriptome due to the loss of RBPm1 (Fig. 4A). To purify the RBPm1-null male gametocytes for comparison, we deleted *Rbpm1* in the *DFsc7* line. The mutant line *DFsc7;*Δ*Rbpm1* displayed the same phenotypes as Δ*Rbpm1* (Supplementary Fig. 3A–D). Purified male gametocytes of the *DFsc7;*Δ*Rbpm1* were collected by fluorescence-activated cell sorting for RNA-seq (Supplementary Fig. 3E, F). We analyzed the differentially

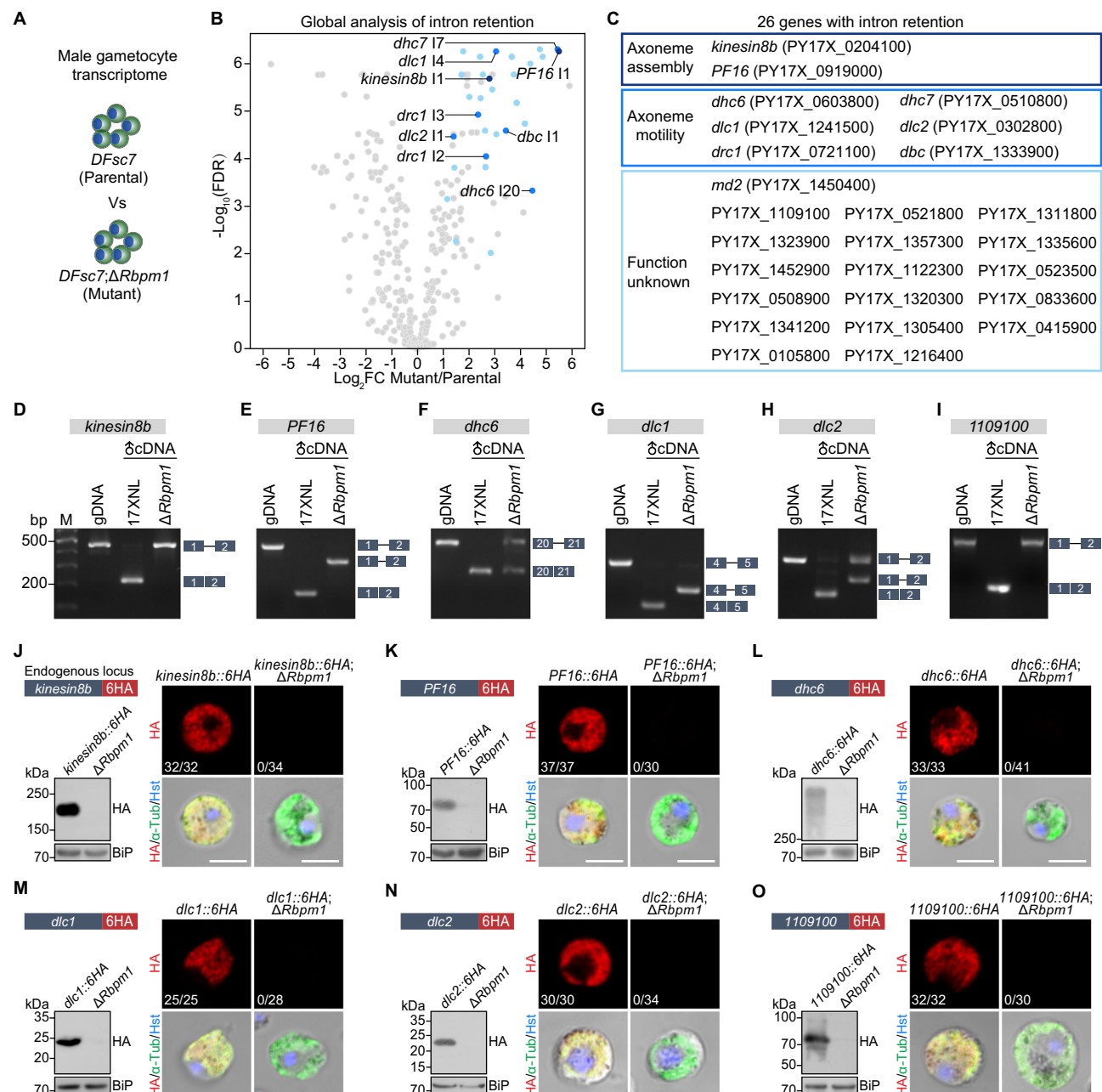

**Fig. 4 | RBPm1 deficiency causes intron retention and protein loss of axonemal genes. A** A schematic showing the transcriptome analysis of RBPm1-null male gametocytes. *DFsc7;ΔRbpm1* is a *DFsc7*-derived RBPm1 mutant line. *DFsc7* (parental) and *DFsc7;ΔRbpm1* (mutant) male gametocytes were sorted by FACS for RNA-seq. **B** Global analysis of differential intron retention identified 30 retained introns (blue dots) in the mutant versus the parental line. Retained introns with $\log_2 FC \geq 1$ and $p \leq 0.05$ were further verified manually by visualization in IGV. The *p*-values were calculated by quasi-likelihood F-test and adjusted by FDR. These introns were originated from 26 genes. Detailed information of these genes and introns is provided in Supplementary Fig. 4A. **C** List of the 26 genes with intron retention from (**B**), categorized by protein function. **D–I** RT-PCR confirmation of intron retention in 6 selected genes *kinesin8b*, *PF16*, *dhc6*, *dlc1*, *dlc2*, and *PY17X_1109100*. Genomic DNA (gDNA) from 17XNL parasite, complementary DNA (cDNA) from male gametocytes of parental and mutant parasites were analyzed. Exons are indicated by boxes and introns by lines. Three independent experiments with similar results. RT-PCR analysis for all 26 genes is presented in Supplementary Fig. 5A. **J–O**. Protein expression analysis of the 6 genes shown in (**D–I**) in male gametocytes after loss of RBPm1. Each endogenous gene was tagged with a 6HA at the C-terminus in both 17XNL and *ΔRbpm1* parasites (the schematic in the top left panel), generating two tagged lines. Immunoblot of the 6HA-tagged protein in gametocytes with and without RBPm1 (bottom left panel). IFA of the 6HA-tagged protein in male gametocytes with and without RBPm1 (right panel). x/y at bottom-left represents the number of HA-positive male gametocytes/the total number of male gametocytes tested. Three independent experiments with similar results. Scale bars: 5 μm.

expressed genes between *DFsc7* and *DFsc7;ΔRbpm1* (Supplementary Data 2). As expected, the *Rbpm1* transcripts were undetectable in the *DFsc7;ΔRbpm1* (Supplementary Fig. 3G, H). RBPm1 deficiency led to changed expression of several genes (Supplementary Fig. 3G), but none of the differentially expressed genes was known to be implicated in axoneme assembly during male gametogenesis.

We found 30 intron retention (IR) events in transcripts of 26 genes after loss of RBPm1 (Fig. 4B, C) by bioinformatic analysis of global intron retention and manual examination on Integrative Genomics Viewer[38]. These genes were specifically or preferentially transcribed in the male gametocytes (Supplementary Fig. 4A). Among them (Fig. 4C), the orthologs of *kinesin8b* and *PF16* had been reported essential for

axoneme assembly of male gametogenesis in *P. berghei*[39–41]. Six putative dynein motor-associated genes, *dhc6* (dynein heavy chain, PY17X_0603800), *dhc7* (dynein heavy chain, PY17X_0510800), *dlc1* (dynein light chain, PY17X_1241500), *dlc2* (dynein light chain, PY17X_0302800), *drc1* (dynein regulatory complex protein, PY17X_0721100), and *dbc* (dynein beta chain, PY17X_1333900), were included. The *md2* (male development protein 2, PY17X_1450400), a male gene recently identified[42], was also included. The rest 17 IR genes had not been previously described in the *Plasmodium*. Gene Ontology (GO) enrichment analysis of these IR genes found significant GO terms that are associated with MT or cytoskeleton (Supplementary Fig. 4B). RT-PCR using the primers anchored in the flank exons of each of 26 introns further confirmed that these introns were retained in the transcripts in the absence of RBPm1, while their neighboring introns were correctly removed (Fig. 4D–I and Supplementary Fig. 5A). Using RT-qPCR, we further confirmed the IR of *kinesin8b* intron1 and *PF16* intron1 in the RBPm1-null male gametocytes (Supplementary Fig. 5B). Interestingly, the whole part of intron was retained in the transcripts for most IR genes, while only a N-terminal part of intron was retained for three IR genes, including *PF16* intron1, *dlc1* intron4, and PY17X_1311800 intron5 (Fig. 4D–I and Supplementary Fig. 5A). Therefore, RBPm1 is required for the splicing of selective introns in certain male genes, especially MT or cytoskeleton-related genes.

We speculated that the RBPm1-regulated IR genes are axonemal given the following facts: (1) RBPm1 depletion causes defective axoneme assembly; (2) All IR genes are male-specific; (3) 8 IR genes are axoneme-related. To test it, we selected 12 out of the 26 genes, including 6 annotated (*kinesin8b*, *PF16*, *dhc6*, *dhc7*, *dlc1*, *dlc2*) and 6 unannotated (PY17X_1109100, PY17X_0521800, PY17X_1311800, PY17X_1323900, PY17X_1357300, PY17X_1335600). Each gene was endogenously tagged at the N- or C-terminus with a 6HA in the 17XNL. All 12 proteins were specifically expressed in male gametocytes during parasite life cycle (Supplementary Fig. 6A), in agreement with their transcript profile. In the inactivated gametocytes, these proteins were distributed in the cytoplasm, while after activation, 11 of 12 proteins displayed axoneme localization in the flagellating male gametes (Supplementary Fig. 6B–M). These results suggested that RBPm1 controls intron splicing for a group of the axonemal genes.

## Intron retention leads to loss of axonemal protein in RBPm1-null male gametocytes

Nucleotide sequence analysis revealed that IR would result in premature translation and thus cause loss of protein expression for the axonemal genes (Supplementary Fig. 7). To analyze the effect of IR on the axonemal proteins after RBPm1 loss, we deleted *Rbpm1* gene in each of two tagged lines *kinesin8B::6HA* and *PF16::6HA* (Fig. 4J, K). In the absence of RBPm1, 6HA-tagged Kinesin8B, and PF16 were not detected or under detectable thresholds in male gametocytes compared to the parental counterparts in both IFA and immunoblot (Fig. 4J, K). To further confirm the protein loss, we analyzed 4 other IR genes *dhc6*, *dlc1*, *dlc2*, and PY17X_1109100. Endogenous *Rbpm1* gene was deleted in all the 4 tagged lines (*dhc6::6HA*, *dlc1::6HA*, *dlc2::6HA*, and *1109100::6HA*) (Fig. 4L–O). These 6HA-tagged proteins lost expression in the RBPm1-null male gametocytes (Fig. 4L–O), similarly as Kinesin8B and PF16 did. These results demonstrated that RBPm1 deficiency causes expression loss of target axonemal proteins.

To confirm the essential roles of *P. yoelii* Kinesin8B and PF16 in axoneme assembly as reported in *P. berghei*[39–41], we disrupted *kinesin8b* and *PF16* genes in the 17XNL, obtaining mutant lines Δ*kinesin8b* and Δ*PF16* (Supplementary Fig. 8A). As expected, depletion of *kinesin8b* or *PF16* either blocked or severely impaired male gamete formation, respectively (Supplementary Fig. 8B). Neither mutant produced any midgut oocysts in the infected mosquitoes (Supplementary Fig. 8C). Ultrastructure analysis of male gametocytes at 8 mpa revealed that the Δ*kinesin8b* mutant failed to develop "9 + 2" axoneme,

with loss of both central and peripheral MTs, while most of the axonemes lost central MTs (shown as "9 + 0" or "9 + 1") in the Δ*PF16* (Supplementary Fig. 8D, E), in line with gene disruption phenotypes in *P. berghei*[39–41]. Therefore, depletion of Kinesin8B or PF16 phenocopies RBPm1 deficiency in axoneme assembly.

## Intron deletion restores axonemal proteins and partially rectifies axoneme assembly defects in RBPm1-null gametocytes

Since IR disrupted the axonemal proteins expression, we tested whether enforced genomic deletion of the retained intron could restore protein expression by bypassing intron splicing at the transcripts in RBPm1-null male gametocytes. The endogenous *kinesin8b* intron1 (239 bp) was removed in the *kinesin8b::6HA*;Δ*Rbpm1* parasite by CRISPR-Cas9 (Fig. 5A), generating the intron-null mutant *kinesin8b*Δ*intron1* (*kinesin8b*Δ*I1*). Both IFA and immunoblot revealed that the deletion of intron1 restored Kinesin8B::6HA expression to WT level in the RBPm1-null gametocytes (Fig. 5B, C). To further confirm the restoration effect, we tested 3 other retained introns (*PF16* intron1, *dlc1* intron4, and PY17X_1109100 intron1). Compared to the parental RBPm1-null parasites, the expression of PF16::6HA and 1109100::6HA in male gametocytes were fully restored (Fig. 5D–F, J–L), while the Dlc1::6HA was partially restored after removal of the corresponding intron (Fig. 5G–I). Expression restoration of these axonemal proteins (Kinesin8b, PF16, Dlc1, and PY17X_1109100) via intron deletion strongly confirmed the causative effect of IR on axonemal protein loss in the absence of RBPm1.

We next tested whether genomic deletion of the retained introns could rescue or rectify the defective axoneme assembly in the Δ*Rbpm1* mutant. We deleted the *kinesin8b* intron1 in the Δ*Rbpm1* line, but this deletion of single intron failed to restore any EC formation in the Δ*Rbpm1;kinesin8b*Δ*intron1* parasites. However, compared to complete lack of axonemes showing "9 + 2", "9 + 1", or "9 + 0" MTs in the parental Δ*Rbpm1*, some axoneme-like structures ("9 + 2": 1%, "9 + 1": 3%, and "9 + 0": 15%) were detected in the Δ*Rbpm1;kinesin8b*Δ*intron1* (Fig. 5M, N), indicating that deletion of *kinesin8b* intron1 could partially rescue the defective axoneme assembly caused by RBPm1 deficiency. Notably, additional deletion of the *PF16* intron1 in the Δ*Rbpm1;kinesin8b*Δ*intron1* parasite further mitigated axoneme defects in the resulted Δ*Rbpm1;kinesin8b*Δintron1;*PF16*Δintron1 parasite line ("9 + 2": 1%, "9 + 1": 11%, and "9 + 0": 32%) (Fig. 5M, N). These results demonstrated that RBPm1 regulates axoneme assembly by controlling intron splicing of a group of axonemal genes. Without RBPm1, deletion of 2 introns (*kinesin8b* intron1 and *PF16* intron1) was insufficient to restore axoneme assembly to the WT level ("9 + 2": 93%) (Fig. 5M, N). Therefore, in addition to *kinesin8b* and *PF16*, other axonemal genes targeted by RBPm1 may also play important roles in axoneme assembly during male gametogenesis.

## RBPm1 interacts with spliceosome E complex and introns of axonemal genes

To investigate whether RBPm1 associates with the spliceosome responsible for intron splicing, we used the biotin ligase TurboID-based proximity labeling to identify RBPm1-interacting proteins in the gametocytes. The endogenous RBPm1 was tagged with a HA::TurboID motif in the 17XNL, generating the line *Rbpm1::TurboID* (Fig. 6A). A control parasite *Rbpm1::T2A::TurboID* was generated by fusing endogenous RBPm1 with a "ribosome skip" T2A peptide, a NLS (nuclear localization signal), and a HA::TurboID (Fig. 6A), permitting separated expression of RBPm1 and biotin ligase. Gametocytes expressing the ligase were incubated with 50 μM biotin for 20 min at 37 °C. Staining with fluorescent-conjugated streptavidin and anti-HA antibody detected a nuclear distribution of biotinylated proteins in both TurboID-modified gametocytes (Supplementary Fig. 9A), indicating biotinylation of the potential RBPm1-interacting proteins in the nucleus. Mass spectrometry of the streptavidin affinity purified proteins from the

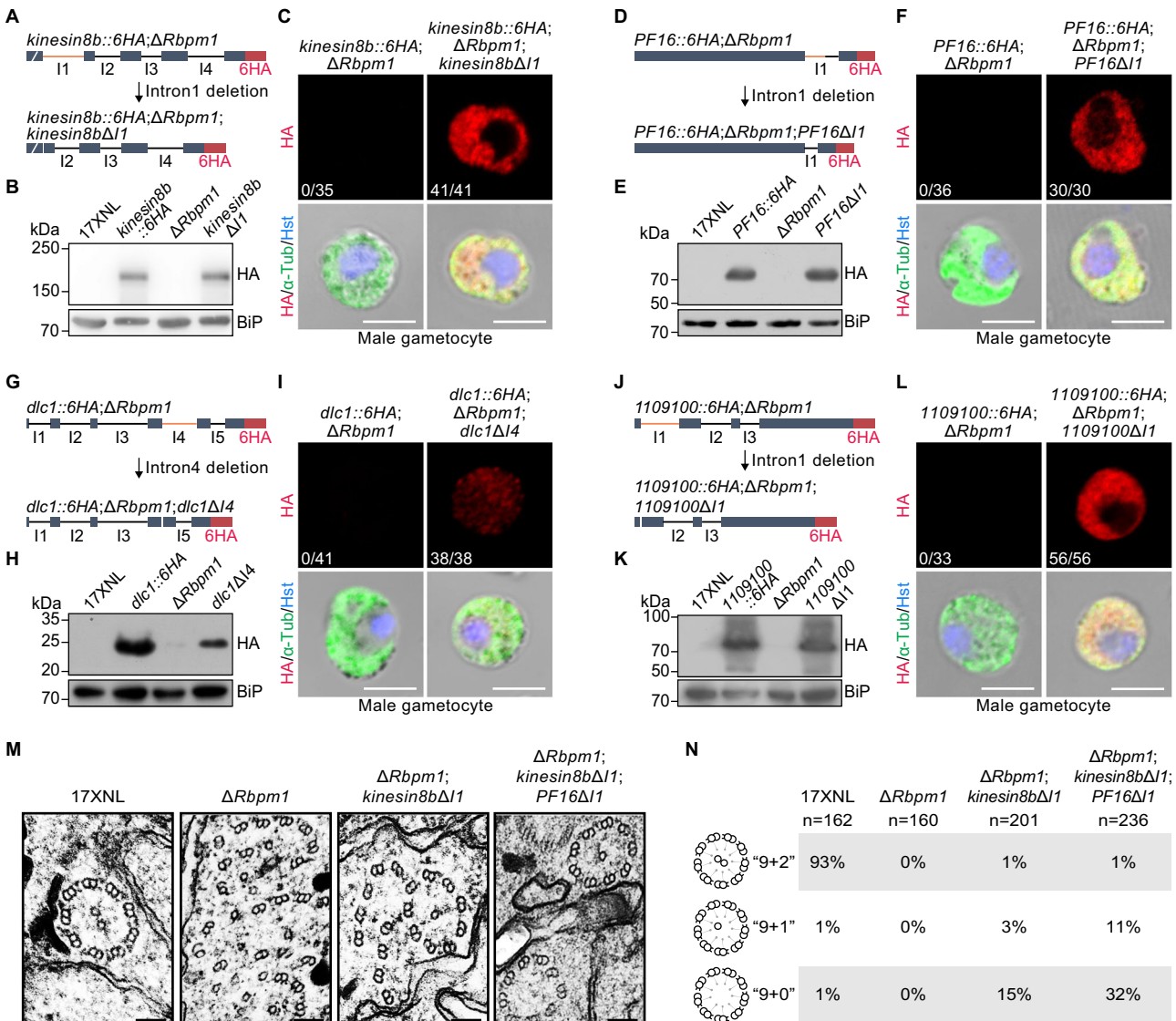

**Fig. 5 | Intron deletion restores axonemal protein expression and partially rectifies axoneme assembly defects in RBPm1-null male gametocytes. A** A schematic showing the genomic deletion of retained intron (*kinesin8b* intron1, orange) in the *kinesin8b::6HA;ΔRbpm1* parasite (abbreviated as *ΔRbpm1*), generating the mutant *kinesin8b::6HA;ΔRbpm1;kinesin8bΔintron1* (abbreviated as *kinesin8bΔI1*). **B** Immunoblot of 6HA-tagged Kinesin8B protein in gametocytes. Three independent experiments with similar results. **C** IFA of 6HA-tagged Kinesin8B in male gametocytes. x/y represents the number of HA-positive male gametocytes/the total number of male gametocytes tested. Three independent experiments with similar results. Scale bars: 5 μm. **D, E, F** Effect of intron deletion (*PF16* intron1) on the restoration of PF16 protein in RBPm1-null male gametocytes. Similar analysis as in

(**A**, **B**, **C**). **G, H, I** Effect of intron deletion (*dlc1* intron4) on the restoration of Dlc1 protein in RBPm1-null male gametocytes. **J, K, L** Effect of intron deletion (*PY17X_1109100* intron1) on the restoration of PY17X_1109100 protein in RBPm1-null male gametocytes. **M** Transmission electron microscopy of axoneme architecture in male gametocytes at 8 mpa. *ΔRbpm1;kinesin8b*ΔI1 is a *ΔRbpm1*-derived modified line with deletion of *kinesin8b* intron1. *ΔRbpm1;kinesin8b*ΔI1;*PF16*ΔI1 is a *ΔRbpm1* derived modified line with deletion of both *kinesin8b* intron1 and *PF16* intron1. Scale bars: 100 nm. **N** Quantification of axoneme formation from parasites in (**M**). n is the total number of the intact and defective axoneme structures observed in each group. Three independent experiments with similar results.

*Rbpm1::TurboID* resulted in a list of 113 proteins enriched with high confidence compared to the control (Fig. 6B and Supplementary Data 3). RBPm1 was the top hit, confirming cis-biotinylation of RBPm1 (Fig. 6B). Among the significantly enriched proteins, we found the components of the spliceosome earliest assembling E complex[43–45], including the U1 small nuclear ribonucleoproteins (snRNP) U1-70K, U1-A, U1-C, Sm-B, Sm-D1, Sm-D2, Sm-D3, Sm-E, Sm-F and Sm-G (Fig. 6B, C), and three E complex key factors SF1, U2AF1, and U2AF2 (Fig. 6B, C). Tagging the endogenous U1-70K, U1-A, and U1-C proteins with 4Myc in the *Rbpm1::6HA* parasite showed that these three U1 snRNPs co-localized with RBPm1 in the nucleus (Fig. 6D). Co-immunoprecipitation also confirmed the interaction between RBPm1 and these U1 snRNPs (Fig. 6E–G). Spliceosome A, B, and C complex are formed after the

assembly of splicing initiating E complex[46,47]. However, the components of A, B, and C complex were not detected (Supplementary Fig. 9B, C). Therefore, RBPm1 interacted only with spliceosome E complex, possibly helping to initiate splicing for certain introns in the axonemal genes (Fig. 6H).

Nuclear localization and interaction with spliceosome E complex imply that RBPm1 may bind to the target introns in the pre-mRNA of axonemal genes. We performed UV crosslinking RNA immunoprecipitation (UV-RIP) followed by RT-qPCR with primers recognizing the target pre-mRNAs. In the *Rbpm1::6HA* gametocytes, RBPm1 bound to the intron1 of the *kinesin8b* transcripts using anti-HA nanobody (Fig. 6I). As a control, RIP using anti-GFP nanobody detected no binding (Fig. 6I). Additionally, we analyzed the interaction between

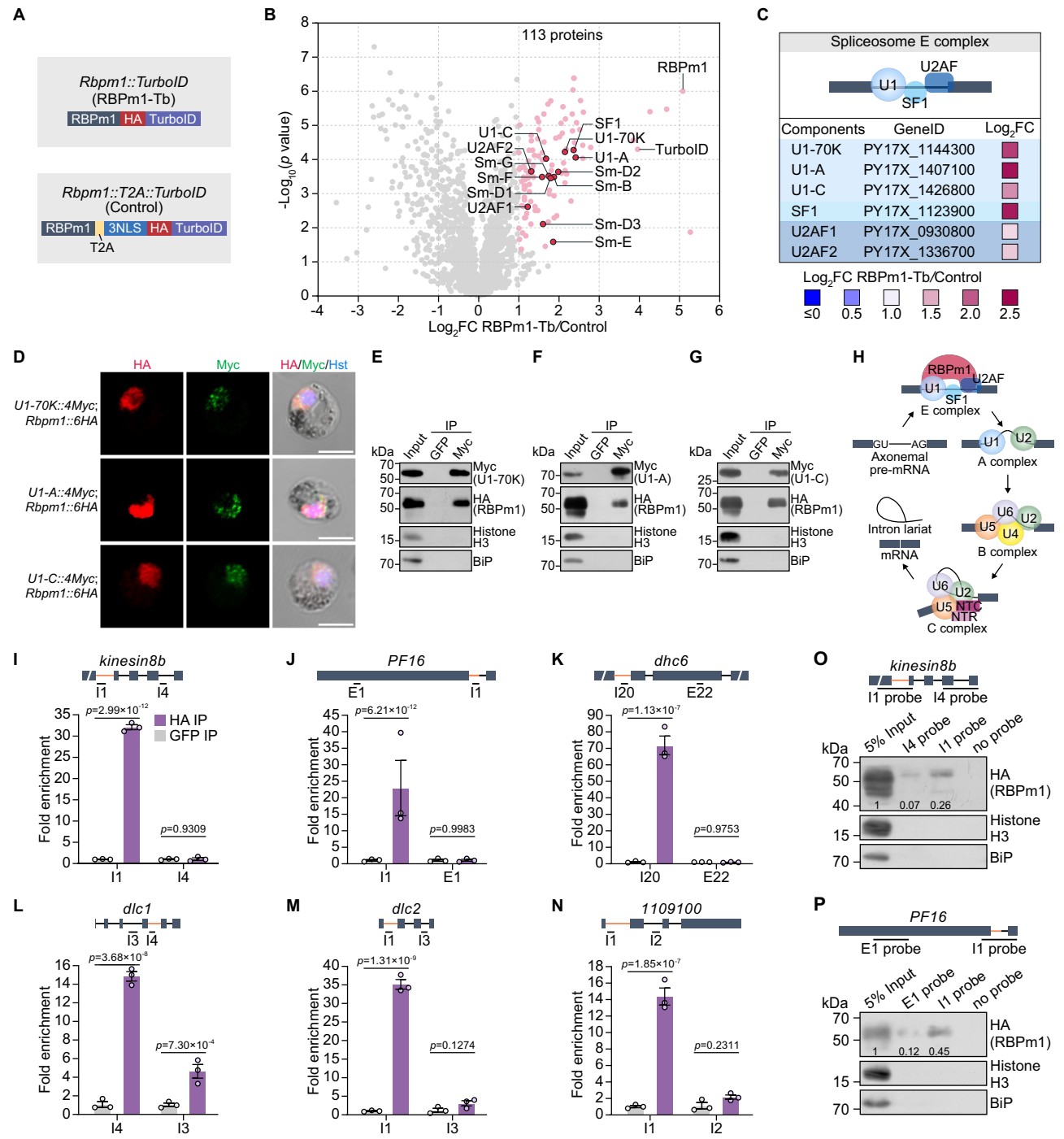

RBPm1 and five other target introns (*PF16* intron1, *dhc6* intron20, *dlc1* intron4, *dlc2* intron1, and *PY17X_1109100* intron1). As expected, RBPm1 bound these target introns. As expected, RBPm1 bound these target introns but not the neighboring introns or exons since each intron is individually excised as a lariat RNA during the splicing (Fig. 6J–N).

Furthermore, we used RNA pull-down to validate the interaction of RBPm1 with the *kinesin8b* intron1 and *PF16* intron1. A biotinylated 500nt RNA probe *kinesin8b* I1 and a control probe *kinesin8b* I4 were synthesized (Fig. 6O, upper panel) and incubated with the *Rbpm1::6HA* gametocyte lysate. The potential RNA-interacted proteins were precipitated using the streptavidin beads and detected by immunoblot. The *kinesin8b* I1 probe retrieved more RBPm1 protein than the *kinesin8b* I4 probe (Fig. 6O). Similarly, the *PF16* I1 probe captured more RBPm1 protein than the probe *PF16* E1 (Fig. 6P). Both RIP and RNA pull-

down experiments supported that RBPm1 binds the *kinesin8b* intron1 and *PF16* intron1.

## RBPm1 directs splicing of axonemal introns inserted in a reporter gene

To further investigate the interaction between RBPm1 and the axonemal introns, we test whether RBPm1 could direct splicing of target introns when inserted into a reporter gene. We developed a blue fluorescence protein (BFP) reporter assay that allows an easy splicing readout in male and female gametocytes of the *DFsc7* parasite. The intact *bfp* transcript driven by the *hsp70* 5′-UTR and the *dhfr* 3′-UTR was integrated into the *p230p* locus of *DFsc7* using CRISPR-Cas9, generating the control line *BFP* (Fig. 7A). The *kinesin8b* intron1 (*Kin8b*I1, 239 bp) was inserted to the *bfp* gene at the nucleotides

**Fig. 6 | RBPm1 interacts with spliceosome E complex and introns of axonemal genes. A** A schematic showing two modified parasite lines generated for searching RBPm1-interacting proteins by TurboID-based proximity labeling and mass spectrometry. The motif of HA::TurboID and T2A::3NLS::HA::TurboID, respectively, was inserted at the C-terminus of the endogenous RBPm1, generating the line *Rbpm1::TurboID* and the control line *Rbpm1::T2A::TurboID*. **B** Volcano plot displaying 113 significantly enriched proteins (pink dot, cutoffs log$_2$FC ≥ 1 and $p \leq 0.05$) in the *Rbpm1::TurboID* versus *Rbpm1::T2A::TurboID*. Among them, 13 sub-units (red dot) of the early spliceosome E complex were included. The *p*-values were calculated by two-sided *t*-test and adjusted by FDR. **C** Protein interaction analysis between RBPm1 and six spliceosome E complex subunit proteins (U1-70K, U1-A, U1-C, SF1, U2AF1, and U2AF2) from (**B**). **D** IFA of 6HA-tagged RBPm1 and 4Myc-tagged U1 snRNP proteins (U1-70K, U1-A, and U1-C) in male gametocytes of three double-tagged parasites. Three independent experiments with similar results. Scale bars: 5 μm. **E** Co-immunoprecipitation of RBPm1 and U1-70K in gametocytes of the double-tagged parasite *U1-70K::4Myc;Rbpm1::6HA*. Anti-Myc nanobody was used. Bip as a loading control. Three independent experiments with similar results. **F** Co-immunoprecipitation of RBPm1 and U1-A in gametocytes of the double-tagged parasite *U1-A::4Myc;Rbpm1::6HA*. Three independent experiments with similar results. **G** Co-immunoprecipitation of RBPm1 and U1-C in gametocytes of the double-tagged parasite *U1-C::4Myc;Rbpm1::6HA*. Three independent experiments

with similar results. **H** Proposed model showing the interaction between RBPm1 and early spliceosome E complex for intron splicing of axonemal genes. **I–N**. UV-RIP detection of RBPm1 interaction with the retained introns of 6 axonemal genes (*kinesin8b*, *PF16*, *dhc6*, *dlc1*, *dlc2*, and *PY17X_1109100*). A top schematic shows the exon-intron structure of the RBPm1 target axonemal genes. The retained introns are indicated with orange lines, and the genomic regions for qPCR amplicon are shown. UV-RIP was performed in *Rbpm1::6HA* lines using anti-HA nanobody. Anti-GFP nanobody was used as a control. Bound RNA was analyzed by RT-qPCR. Means ± SEM from three independent experiments, two-sided *t*-test. **O** RNA pull-down assay detecting RBPm1 interaction with *kinesin8b* intron1. A top schematic shows the exon-intron structure of the *kinesin8b* gene. The retained introns are indicated in orange lines. A biotinylated 500 nt RNA probe I1 (comprising intron1 and its flanking sequences) and a control probe I4 (comprising intron4 and its flanking sequences) were used. Proteins via RNA pull-down were immunoblot with anti-HA antibody. The numbers are the relative intensities of bands in the blot. Histone H3 and Bip were used as negative controls. Two independent experiments with similar results. **P** RNA pull-down assay detecting RBPm1 interaction with *PF16* intron1. A top schematic shows the exon-intron structure of the *PF16* gene. The retained introns are indicated in orange lines. A biotinylated 500 nt RNA probe I1 (comprising intron1 and its flanking sequences) and a control probe E1 in exon1 were used. Two independent experiments with similar results.

396–397, generating the line *BFP-Kin8b*I1 (Fig. 7B). The inserted *kinesin8b* intron1 would result in premature translation of the *bfp* transcript if it is not spliced. In the control *BFP* line, BFP was expectedly detected in both male (GFP+) and female (mCherry+) gametocytes (Fig. 7A). However, in the *BFP-Kin8b*I1 line, BFP was detected only in male gametocytes (Fig. 7B), indicating that splicing of *kinesin8b* intron1 in the *bfp* transcripts occurred only in male gametocytes. To prove that splicing of *kinesin8b* intron1 in male was RBPm1-dependent, we deleted *Rbpm1* in the *BFP-Kin8b*I1 line and obtained the mutant line *BFP-Kin8b*I1;Δ*Rbpm1* (Fig. 7C). RBPm1 deletion disrupted BFP expression in the *BFP-Kin8b*I1;Δ*Rbpm1* male gametocytes (Fig. 7C). We parallelly analyzed the *kinesin8b* intron2 (*Kin8b*I2, 148 bp), whose splicing from the native gene transcript required no RBPm1 (Supplementary Fig. 5A). In both transgenic line *BFP-Kin8b*I2 (Fig. 7D) and its derivative mutant line *BFP-Kin8b*I2;Δ*Rbpm1* (Fig. 7E), BFP was detected in both male and female gametocytes, confirming RBPm1-independent splicing of *kinesin8b* intron2 from the *bfp* transcript.

Using the reporter assay, we tested 3 other target introns, including *PF16* intron1 (Fig. 7F, G), *dlc1* intron4 (Supplementary Fig. 10A, B), and *PY17X_1109100* intron1 (Supplementary Fig. 10C, D). The *PF16* intron1 (276 bp) was inserted to the *bfp* at the nucleotides 500-501 (Fig. 7F), the *dlc1* intron4 (193 bp) at the nucleotides 455–456 (Supplementary Fig. 10A), while the *PY17X_1109100* intron1 (353 bp) at the nucleotides 390-391 (Supplementary Fig. 10C). As expected, these introns were spliced from the *bfp* transcript only in male gametocytes (Fig. 7F, Supplementary Fig. 10A, C). Similarly, these introns were not spliced at male gametocytes in the RBPm1-null parasites compared to their parental parasites (Fig. 7G, Supplementary Fig. 10B, D). Additionally, we analyzed the *PY17X_1109100* intron2 (272 bp), which could be spliced in the RBPm1-null parasites (Supplementary Fig. 5A), and found that RBPm1 was not required for splicing of this intron from *bfp* transcript in both male and female gametocytes (Supplementary Fig. 10E, F).

Furthermore, we analyzed the RBPm1 interaction with the *kinesin8b* intron1 and *PF16* intron1 in the *bfp* transcript by RNA pull-down. A biotinylated RNA probe *bfp-Kin8b*I1, corresponding to the *kinesin8b* intron1-inserted *bfp* transcript (Fig. 7H, upper panel), retrieved significantly more RBPm1 from the *Rbpm1::6HA* gametocyte lysate compared to the control probe *bfp* (Fig. 7H, middle panel). Similarly, the probe *bfp-PF16*I1, corresponding to the *PF16* intron1-inserted *bfp* transcript, captured more RBPm1 than the control probe *bfp* (Fig. 7H, lower panel). Therefore, RBPm1 could recognize the axonemal introns in the reporter transcript for splicing (Fig. 7I).

## RBPm1 directs splicing of axonemal introns inserted in an endogenous gene

In addition to the reporter gene, we also tested whether RBPm1 could direct splicing of target introns when inserted into an endogenous gene which does not require RBPm1 for intron splicing. We chose the *gep1*, a 4-exon gene expressed in both gender gametocytes and essential for initiating both genders' gametogenesis[48]. We analyzed male and female gametogenesis by measuring EM rupture (TER119 staining), genome replication (DNA staining), and axoneme assembly (α-Tubulin staining). Compared to 17XNL, the *gep1*-deleted parasite line Δ*gep1* expectedly lost ability in EM rupture, genome replication, and axoneme assembly in activated male gametocytes, as well as EM rupture in activated female gametocytes (Fig. 8A, B, E, F, G, and H). Using CRISPR-Cas9, the *kinesin8b* intron1 was inserted into the exon3 of *gep1* locus at the nucleotides 273-274 in the 17XNL (Fig. 8C), while the *PF16* intron1 inserted into the exon1 at the nucleotides 885-886 (Fig. 8D). In both intron-inserted lines *gep1-Kin8b*I1 and *gep1-PF16*I1, normal male gametogenesis and defective female gametogenesis was speculated because GEP1 is not expressed in female due to no RBPm1-mediated intron splicing from the *gep1* transcript. Notably, both the *gep1-Kin8b* I1 and *gep1-PF16* I1 parasites underwent EM rupture only in male (Fig. 8C–F). These results supported that both *kinesin8b* intron1 and *PF16* intron1 were spliced from the *gep1* transcript only in male gametocytes with RBPm1 expression. Consistent with GEP1 expression in male gametocytes, normal genome replication and axoneme assembly were detected during male gametogenesis in both *gep1-Kin8b* I1 and *gep1-PF16* I1 parasites (Fig. 8C, D, G, H). Collectively, the results from the reporter and the endogenous gene assays (Fig. 8I) indicated that the tested axonemal introns themselves could be specifically recognized by RBPm1 for splicing.

## Intron retention prevents expression of axonemal proteins in female gametocytes

Despite the male-biased transcription, the axonemal genes still displayed low-level transcripts in female gametocytes (Supplementary Fig. 11A, C, E). However, no axonemal proteins are expressed in female gametocytes (Supplementary Fig. 11B, D, F). The facts of no protein product of low-level transcripts for the axonemal genes observed in this study are consistent with results from previous transcriptomic and proteomic studies[30,31,49,50], suggesting a post-transcription regulation for the axonemal genes in female gametocytes. Enforced genomic deletion of the retained intron could restore expression of the axonemal proteins Kinesin8b, PF16, Dlc1,

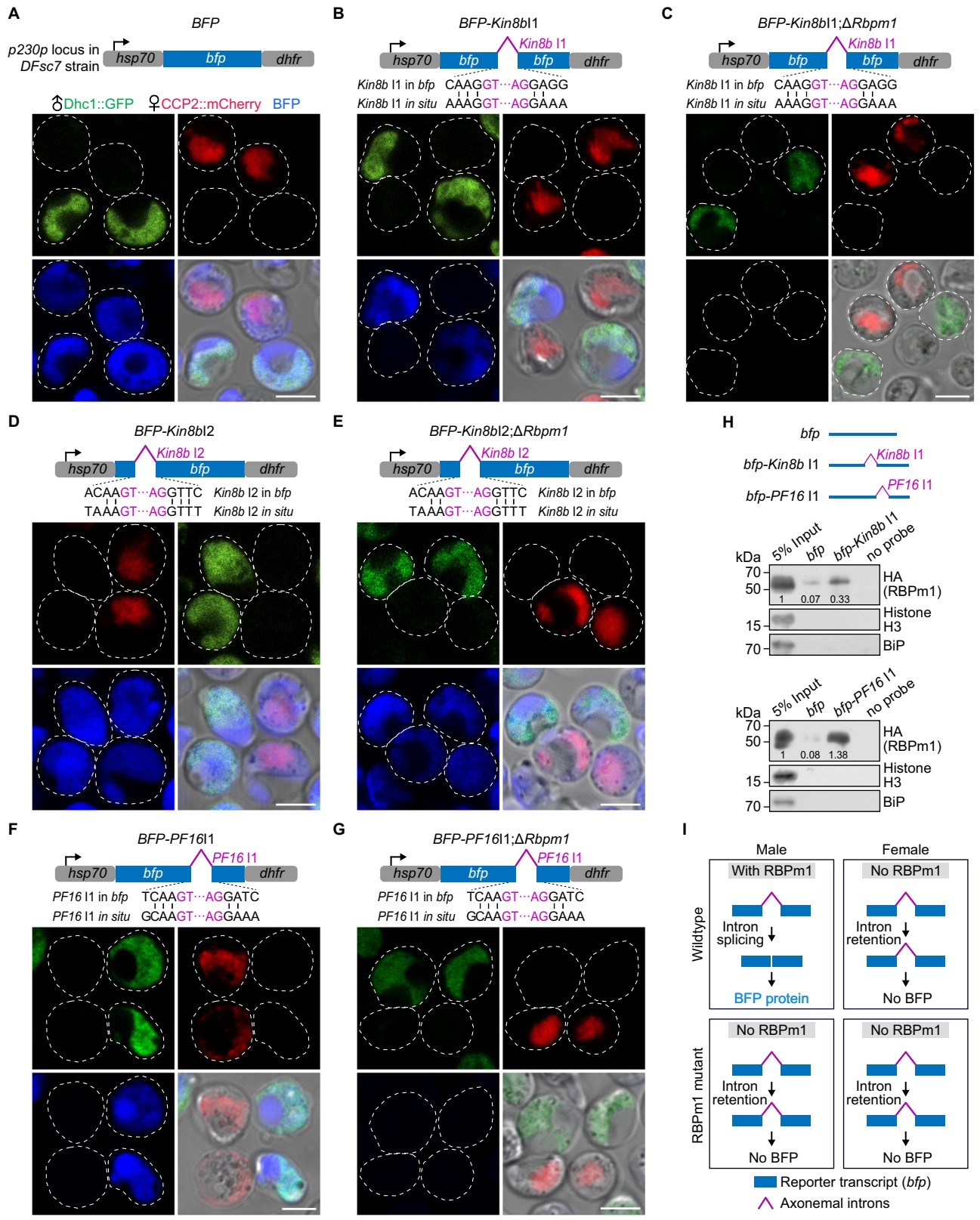

and PY17X_1109100 in the RBPm1-null male gametocytes (Fig. 5A–L). Strikingly, we found that deletion of these introns (*PF16* intron1, *dlc1* intron4, and *PY17X_1109100* intron1) unexpectedly resulted in low-level expression of PF16, Dlc1, and PY17X_1109100 in the counterpart female gametocytes (Supplementary Fig. 11B, D, F). Kinesin8B was not detected in female gametocytes after deletion of *kinesin8b intron1* (Supplementary Fig. 11H), fitting with the extremely low

transcription of *kinesin8b* in female gametocytes (Supplementary Fig. 11G). Furthermore, RT-PCR not only detected the transcripts of these axonemal genes (*PF16*, *dlc1*, and *PY17X_1109100*), but also IR in these transcripts from the purified female gametocytes (Supplementary Fig. 11I–L). These results indicated a role of the RBPm1-target introns in preventing the expression of axonemal proteins in female gametocytes.

**Fig. 7 | RBPm1 directs splicing of axonemal introns inserted in a reporter gene.**
**A** A top schematic shows a transgenic line *BFP* with a *bfp* reporter expression
cassette integrated at the *p230p* locus of the *DFsc7* reporter line. The intact *bfp* is
driven by the 5′UTR of *hsp70* and the 3′UTR of *dhfr*, allowing expression of BFP in
both male (GFP+) and female (mCherry+) gametocytes. Live cell imaging was
shown. Three independent experiments with similar results. Scale bars: 5 μm. **B.** A
transgenic line *BFP-Kin8b*I1 with a *kinesin8b* intron 1 (*Kin8b* I1, purple line)-inserted
*bfp* cassette integrated at the *p230p* locus of the *DFsc7* line. *Kin8b* I1 (purple) was
inserted into the *bfp* gene at the nucleotides 396-397 to mimic the splice site
(vertical lines) of in situ *Kin8b* I1. BFP expression was detected specifically in male
gametocytes of the *BFP-Kin8b*I1 parasites. Three independent experiments with
similar results. Scale bars: 5 μm. **C** A *BFP-Kin8b*I1 derived RBPm1 mutant line *BFP-
Kin8b*I1;*ΔRbpm1*. No BFP expression was detected in male gametocytes of the *BFP-
Kin8b*I1;*ΔRbpm1* parasites. Three independent experiments. Scale bars: 5 μm.
**D** Effect of the *kinesin8b* intron 2 (*Kin8b* I2) insertion on the gametocyte expression
of BFP. Similar analysis as in (**B**). BFP expression was detected in both male and
female gametocytes of the *BFP-Kin8b*I2 parasites. **E** A *BFP-Kin8b*I2 derived RBPm1

mutant line *BFP-Kin8b*I2;*ΔRbpm1*. Similar analysis as in (**C**). BFP expression was
detected in both male and female gametocytes of the *BFP-Kin8b*I2;*ΔRbpm1* para-
sites. **F** Effect of the *PF16* intron1 (*PF16* I1) insertion on the gametocyte expression of
BFP. Similar analysis as in (**B**). BFP expression was detected specifically in male
gametocytes of the *BFP-PF16*I1 parasites. **G** A *BFP-PF16*I1 derived RBPm1 mutant line
*BFP-PF16*I1;*ΔRbpm1*. Similar analysis as in (**C**). No BFP expression was detected in
male gametocytes of the *BFP-PF16*I1;*ΔRbpm1* parasites. **H** RNA pull-down assay
detecting RBPm1 interaction with the *Kin8b* I1 and *PF16* I1 -inserted *bfp* transcripts
from *BFP-Kin8b*I1 and *BFP-PF16*I1 gametocytes, respectively. Three biotinylated
RNA probes *bfp*, *bfp-Kin8b*I1 (corresponding to the *Kin8b* I1-inserted *bfp* transcript),
*bfp-PF16*I1 (corresponding to the *PF16* I1-inserted *bfp* transcript) were used. Proteins
via RNA pull-down were immunoblotted with anti-HA antibody. The numbers are
the relative intensities of bands in the blot. Histone H3 and Bip were used as
negative controls. Two independent experiments with similar results. **I** A schematic
of RBPm1-dependent splicing of axonemal introns inserted in the reporter
transcript.

## Discussion

For efficient transmission, the malaria parasites in the vertebrate host
differentiate into sexual precursor gametocytes that are poised to
rapidly activate to fertile gametes upon entering into the mosquito
midgut for fertilization and further development. So far, a limited
number of transcription and epigenetic factors have been identified
during gametocyte and gamete development[51,52]. Among ~180 putative
*Plasmodium* RBPs, about one-third of RBP genes exhibit stage specific
or elevated expression in the gametocyte[32]. From this list of RBPs, we
identified a previously undescribed male-specific nuclear RBP, RBPm1,
which is essential for male gametogenesis and mosquito transmission
of the *Plasmodium*. RBPm1 operates as a stage- and gender-specific
splicing factor for spliceosome assembly initiation and regulates the
protein expression of a group of 26 male genes, most of which are
axoneme-related.

Recent studies had discovered several RBPs playing roles in the
developmental programs of gametocyte and gametes. During
gametocytes development, UIS12 contributes to the development
of gametocytes of both genders[53]. Disrupting *Puf1* led to a reduction
in gametocytes, especially female gametocytes[54], while *ccr4-1* gene
deletion obstructs male gametocyte development[55]. *Puf2* knockout,
on the other hand, promotes male gametocyte development[56]. The
CCCH zinc finger protein MD3 regulates the gametocyte maturation
and male gametocytogenesis[57]. In female gametocytes, the DOZI/
CITH/ALBA translation repressor complex and PUF2 hold the stored
mRNAs for translation repression until their proteins were needed
during the development of post fertilization[28,58]. Additionally, the
CAF1/CCR4/NOT complex also plays a role in safeguarding the
stored mRNAs from degradation. In male gametocytes, two func-
tional RBPs have been identified. The alternative splicing factor SR-
MG promotes the establishment of sex-specific splicing patterns
and knocking it out reduces the formation of male gamete[59]; the
*ZNF4*'s knockout results in deregulation of 473 genes, including
axonemal dynein-related genes[60]. These documented RBPs and
RBPm1 identified in this study may function together at the post-
transcriptional regulation to shape the male transcriptome for
gametocyte and gamete development.

During the manuscript preparation of this study, another work by
ref. 42 demonstrated that deletion of the *Rbpm1* ortholog of *P. berghei*
(PBANKA_0716500, named as *md5*) had no effect on female and male
gametocyte formation, but resulted in male-specific infertility. These
findings are consistent with the defective male gamete formation
phenotype of the *ΔRbpm1* in *P. yoelii* in this study, indicating conserved
function of RBPm1 in the rodent malaria parasites. As *P. falciparum* is
the most lethal human malaria parasite, future studies are worthy to
investigate whether the RBPm1 ortholog in the *P. falciparum* functions
similarly in male gametogenesis.

The RBPm1-deficient parasites showed specific defects in axo-
neme assembly during male gametogenesis (Fig. 3). Axoneme is a MT
cytoskeleton essential for the eukaryotic flagellar motility, consisting
of a central pair of singlet MTs encircled by 9 outer doublet MTs. This
9 + 2 organization of axonemes is highly conserved in the eukaryotes,
including *Plasmodium*[61]. However, the axoneme in *Plasmodium* differs
from that in other model organisms in several aspects[10,62,63]. First, the
biogenesis of axoneme in *Plasmodium* male gametogenesis is extre-
mely fast, taking only 6-8 min to assemble 8 axonemes[9,10]. Second,
location of basal body. In the canonical cilium, the basal body is
localized under the plasma membrane. In *Plasmodium* male gameto-
cytes, the basal bodies are residing at the nuclear membrane[63]. Third,
location for axoneme assembly. The canonical axoneme protrudes
distally from the cell simultaneously when growing from the basal
body. The *Plasmodium* assemblies the axoneme within the cytoplasm,
independent of intraflagellar transport required for cilium
formation[10,62]. Last, each assembled axoneme associates with a haploid
nuclei to progressively protrude from the parasite plasma membrane,
resulting in a free motile flagellum[10]. Mechanisms underlying the
cytoplasmic assembly and exflagellation of axonemes in *Plasmodium*
remain largely unknown, although the involvement of some conserved
basal body and axonemal proteins has been described, including
armadillo repeat protein PF16[41], motor protein Kinesin8B[39,40,64], basal
body proteins SAS4 and SAS6[63,65–67], and radial spoke protein RSP9[68]. It
is possible that the *Plasmodium* had evolved novel mechanisms to
fulfill the requirement for the axoneme. In this study, we identified a
group of 26 male genes targeted by RBPm1. Several known or putative
axoneme-associated genes were included. Importantly, analysis of
endogenous protein localization showed that most of the tested pro-
teins encoded by RBPm1-target genes co-localizing with axoneme
(Supplementary Fig. 6B–M), suggesting their roles in biogenesis,
structure, regulation, or function of axoneme. For future studies, it will
be intriguing to understand the roles of these 26 RBPm1-regulated
genes, especially 17 previously undescribed ones, during male game-
togenesis in the *Plasmodium*.

In the RBPm1 deficient male gametocytes, 30 IR events were
detected in 26 male genes. One intron was retained in each of 22 genes
while two introns were retained in each of 4 other genes, respectively
(Fig. 4B and Supplementary Fig. 4A). Mechanistically, RBPm1 not only
bound to the intron-retained transcripts, but also interacted with
spliceosome E complex. U1 snRNPs recognize and pair with the 5′
splice site of intron. SF1, U2AF1 and U2AF2 form a complex and bind to
branch point, 3′ splice site, and polypyrimidine tract respectively[69,70].
These above factors assemble the E complex as a spliceosome earliest
stage. After that, the spliceosome dynamically releases and recruits
different snRNPs to establish the assembly for further stages, including
spliceosome A, B, and C complex. RBPm1 was detected to interact

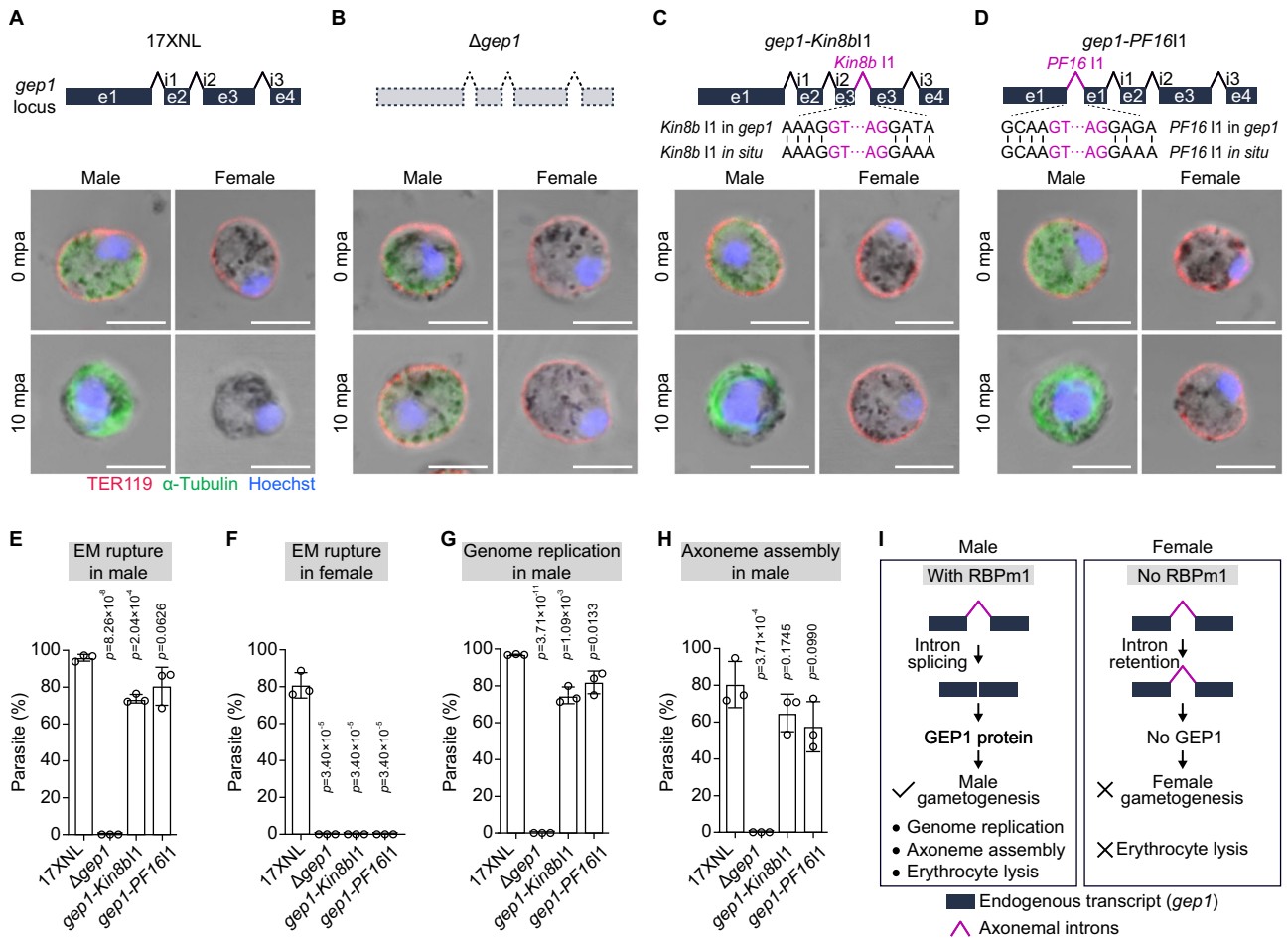

**Fig. 8 | RBPm1 directs splicing of axonemal introns inserted in the endogenous gene. A** A top schematic shows the genomic locus of a 4-exon gene *gep1*, which is expressed in both gender gametocytes and essential for both genders' gametogenesis. Erythrocyte plasma membrane (EM) rupture, genome replication, and cytoplasmic assembly of the axoneme were analyzed in gametocytes of the 17XNL parasites at 10 mpa. The parasites were co-stained with anti-TER-119 and anti-α-Tubulin antibodies and Hoechst 33342. TER-119 (red) negative gametocytes were recognized as EM rupture. Enlarged nuclei represent the genome replication in male gametocytes. Enhanced α-Tubulin (green) signal represents cytoplasmic assembly of the axoneme in male gametocytes. Three independent experiments with similar results. Scale bars: 5 μm. **B.** A top schematic shows a modified line Δ*gep1*, in which the endogenous *gep1* gene was deleted in the 17XNL. Similar analysis for the Δ*gep1* parasites as in (**A**). **C** A top schematic shows a modified line *gep1-Kin8b*I1, in which the *kinesin8b* intron1 (*Kin8b*I1) was inserted into the exon3 of *gep1* locus at the nucleotides 273-274 in the 17XNL. Similar analysis for the *gep1-Kin8b*I1

parasites as in (**A**). **D** A top schematic shows a modified line *gep1-PF16*I1, in which the *PF16* intron1 (*PF16*I1) was inserted into the exon1 of *gep1* locus at the nucleotides 885-886 in the 17XNL. Similar analysis for the *gep1-PF16*I1 parasites as in (**A**). **E** Quantification of EM rupture in male gametocytes of the four parasites tested. Data are means ± SEM from three independent experiments, two-sided *t*-test. **F** Quantification of EM rupture in female gametocytes of the four parasites tested. Data are means ± SEM from three independent experiments, two-sided *t*-test. **G** Quantification of genome replication in male gametocytes of the four parasites tested. Data are means ± SEM from three independent experiments, two-sided *t*-test. **H** Quantification of axoneme assembly in male gametocytes of the four parasites tested. Data are means ± SEM from three independent experiments, two-sided *t*-test. **I** Schematic of RBPm1-dependent splicing of axonemal introns inserted in the endogenous gene *gep1*. Male-specific RBPm1 could recognize and splice the axonemal introns (*kinesin8b* intron1 or *PF16* intron1) inserted in the *gep1* transcript, allowing male-specific GEP expression and thus male gametogenesis.

exclusively with the components of spliceosome E complex, but not those of the A, B, and C complex (Supplementary Fig. 9C). Therefore, RBPm1 likely function as a splicing activator, linking spliceosome E complex with the selective introns of axonemal genes for splice site recognition. In mammals and plants, a RBP of Dek played a similar role and promoted the splicing of certain introns by bridging the intron with the U1/U2 snRNPs[71,72]. At this stage, the data support an association of RBPm1 and spliceosome E complex, but it is not yet clear if it is a direct association.

Both RIP and RNA pull-down assays demonstrated that RBPm1 interacts with the target introns in transcripts of the axonemal genes, suggesting the presence of signals recognized by RBPm1 in these introns. To investigate if the signals for RBPm1 recognition are imparted by the introns themselves but not the adjoining exons, we analyzed the splicing capability of these introns when they were inserted in either a reporter gene (*bfp*) or an irrelevant endogenous

gene (*gep1*). The results from both intron splicing assays establish stringent dependencies of splicing on RBPm1 for these axonemal introns, suggesting intrinsic signals within the introns for RBPm1 recognition. In addition, the adjoining exons may play less modulatory role in the RBPm1 recognition of the axonemal introns. We attempted to search for the common features, such as length, GC content, splice sites, and motif enrichment, but unfortunately observed seemingly no shared features among these 30 RBPm1 target introns. The molecular basis for the axonemal intron recognition by RBPm1 is still unknown. One possibility is that RBPm1 target introns may possess the sequence-independent features, such as RNA structures or epigenetic modifications, for RBPm1 recognition. The structure of RBPm1 is not available yet. To understand the recognition or interaction between RBPm1 and its target introns, future studies into an atomic resolution structure of the protein (RBPm1)-RNA (intron) complex will be required.

Axoneme is an essential cellular structure specifically required for male gametogenesis during the life cycle of *Plasmodium*. Consistent with this physiological requirement of male gametocytes, the axonemal genes display significant male-biased transcription[30,31,49]. Interestingly, these axonemal genes also showed low-level transcripts in female gametocytes in many others and our studies[30,31,49], likely due to the transcription leaking. However, no axonemal proteins are detected in female gametocytes[50], suggesting a post-transcription regulation for the expression turn off of axonemal genes in female gametocytes. We found that genomic deletion of the retained intron (*kinesin8b* intron1, *PF16* intron1, *dlc1* intron4, and *PY17X_1109100* intron1) could bypass the intron splicing and thus restore expression of the axonemal proteins (Kinesin8b, PF16, Dlc1, and PY17X_1109100) in the RBPm1-null male gametocytes. Notably, these introns deletion unexpectedly resulted in low-level expression of PF16, Dlc1, and PY17X_1109100 in female gametocytes (Supplementary Fig. 11B, D, F). The level of proteins restored was correlated with the level of transcripts for these axonemal genes in female gametocytes. These results confirmed the low-level transcripts of these axonemal genes in female gametocytes. In addition, no protein products of these low-level transcripts could be explained by IR and translation failure in female gametocytes. Based on these results, we proposed a dual role of RBPm1-target introns in axonemal gene expression in male and female gametocytes respectively (Supplementary Fig. 11M). In male gametocytes, RBPm1 (as a key)-directed splicing of axonemal intron (as a lock) allows protein expression of axonemal genes for axoneme assembly. In female gametocytes, dual blockage via weak transcription and IR shuts the protein expression of the axonemal genes. The splicing activator RBPm1 and its target introns constitute an intron splicing program, safeguarding the expression of axonemal proteins in male gametocytes while preventing the expression of these proteins in female gametocytes, to fulfill the sexually dimorphic protein profiles during sexual development of the *Plasmodium*.

## Methods

### Animals and ethics statement
The animal experiments conducted in this study were approved by the Committee for Care and Use of Laboratory Animals of Xiamen University (XMULAC20190001). Female ICR mice aged 5–6 weeks were acquired from the Animal Care Center of Xiamen University. The mice were housed in a controlled environment at 22–24 °C, relative humidity of 45–65%, and a 12-h light/dark cycle. They were used for parasite propagation, drug selection, parasite cloning, and mosquito feeding. The larvae of *Anopheles stephensi* mosquitoes (Hor strain) were maintained in an insect facility under controlled conditions of 28 °C, 80% relative humidity, and a 12-h light/12-h dark cycle. Adult mosquitoes were fed with a 10% (w/v) sucrose solution containing 0.05% 4-aminobenzoic acid and kept at 23 °C.

### Plasmid construction
All genetically modified parasites in this study are listed in Supplementary Table 1. The CRISPR/Cas9 plasmid pYCm was used for gene editing[34,73]. To construct plasmids for gene tagging, the 5′- and 3′-flanking sequences (300–700 bp) at the designed insertion site of target genes were amplified as homologous templates. DNA fragments encoding 6HA, GFP, or 4Myc were placed between them and in-frame with the target gene. To construct the plasmids for gene knockout, the left and right homologous arms consisted of 400–700 bp sequences upstream and downstream of the coding sequences of the target gene. To construct the plasmids for domain or intron deletion, the left and right homologous arms consisted of 200–700 bp sequences upstream and downstream of the domain or intron were PCR-amplified and inserted into specific restriction sites in pYCm. To construct the plasmids for intron insertion, the left and right homologous arms were composed of gene genomic sequences ranging from 300–600 bp upstream and downstream of the insertion site, respectively. The left homologous arm, intron, and right homologous arm were connected by overlap PCR, and the fused fragment was inserted into specific restriction sites in pYCm. In each modification, at least two small guide RNAs (sgRNAs) were designed. To construct the plasmids for the *bfp* reporter assay, the intact *bfp* reporter (717 bp) driven by the 5′-UTR (1755 bp) of the *hsp70* gene and the 3′-UTR (561 bp) of the *dhfr* gene were inserted into specific restriction sites between the left and right homologous arms for transgenic integration in the *p230p* locus of *P. yoelii*[74]. The *kinesin8b* intron1 (239 bp), *kinesin8b* intron2 (148 bp), *PF16* intron1 (276 bp), *dlc1* intron4 (193 bp), PY17X_1109100 intron1 (353 bp), and PY17X_1109100 intron2 (272 bp) were inserted into the *bfp* reporter by overlap PCR. All primers and oligonucleotides used in the plasmid construction are listed in Supplementary Table 2.

### Parasite transfection and genotyping
The procedures for parasite transfection and genotyping were carried out as previously described[34,73]. Briefly, the schizonts were isolated from infected mice using a 60% Nycodenz density gradient centrifugation. The parasites were then electroporated with 5 μg plasmid using a Nucleofector 2b Device (Lonza, Germany). The transfected schizonts were immediately intravenously injected into a naïve mouse, and pyrimethamine (Pyr) selection (6 mg/ml in drinking water) was applied the day following transfection. Pyr-resistant parasites were typically observed about 7 days after drug selection. Single clone of parasite was obtained by limiting dilution in mice, and genomic DNA was extracted from infected mouse blood for PCR genotyping using specific primers listed in Supplementary Table 2. PCR confirmation of correct 5′ and 3′ homologous recombination in each gene modification are presented in Supplementary Fig. 12.

### Negative selection with 5-fluorocytosine
To remove the pYCm plasmids, we employed negative selection using 5-fluorocytosine. A mouse infected with the modified parasite clone was given drinking water containing 2 mg/ml of 5-fluorocytosine (Sigma-Aldrich, cat#F6627) in a dark bottle. After ~3 days, most of the surviving parasites no longer carried pYCm plasmids and underwent limiting dilution cloning by injecting into mice via the tail vein. Seven days later, blood smears were used to identify the mice that were infected with parasites, and these parasites were genotyped again and used as the single cloned parasite.

### Gametocyte induction in mice
The ICR mice were treated with phenylhydrazine (80 μg/g body weight; Sangon Biotech, China, cat#A600705-0025) to induce hyper-reticulocytosis. Three days post-treatment, the mice were infected with $4 \times 10^6$ asexual stage parasites via tail vein injection. The peak of gametocytaemia usually occurred on day three post-infection. Male and female gametocytes were counted using Giemsa-stained thin blood films, and gametocytaemia was calculated as a percentage of the number of male or female gametocytes over the number of parasitized erythrocytes.

### Gametocyte purification
The procedures for gametocyte purification were carried out according to previously described methods[48]. Briefly, ICR mice were intra-peritoneally treated with phenylhydrazine 3 days prior to parasite infection. Starting from 2 days post-infection, the mice were orally administered 0.12 mg/d of sulfadiazine (Sigma, cat#S8626) for 2 days to eliminate asexual stage parasites. Approximately 1 ml of mouse blood containing gametocytes was collected from the orbital sinus and then suspended in 6 ml of gametocyte maintenance buffer (GMB). GMB comprises 137 mM NaCl, 4 mM KCl, 1 mM $CaCl_2$, 20 mM glucose, 20 mM HEPES, 4 mM $NaHCO_3$, 0.1% BSA, and has a pH of 7.2. The 7 ml

parasite sample was layered on top of a 2 ml 48% Nycodenz/GMB cushion in a 15 ml centrifugation tube. The cushion consisted of 27.6% w/v Nycodenz in 5 mM Tris·HCl (pH 7.2), 3 mM KCl, and 0.3 mM EDTA. After centrifugation at 1900 g for 20 min, the gametocytes were collected from the interphase and washed twice with GMB for further use.

### Exflagellation assay of male gametocytes

2.5 μl of mouse tail blood with gametocytes was mixed with 100 μl of exflagellation medium. The exflagellation medium was composed of RPMI 1640 supplemented with 100 μM xanthurenic acid (XA, Sigma, cat#D120804), 2 unit/ml heparin, and pH 7.4. The mixture was incubated at 22 °C for 10 min. The number of parasite exflagellation centers (ECs) and total red blood cells were counted within a 1 × 1-mm square area of a hemocytometer under a light microscope. The exflagellation rate was calculated as the number of ECs per 100 male gametocytes.

### In vitro ookinete culture

Mouse blood with the gametocytes was collected in the heparin-containing tubes and immediately mixed with the ookinete culture medium. This medium consisted of RPMI 1640 supplemented with 25 mM HEPES, 10% fetal calf serum, 100 μM XA, and had a pH of 8.0. The blood/medium volume ratio was 1:10. The parasite samples were incubated at 22 °C for 16 h and analyzed using Giemsa-stained thin blood films. The number of ookinetes (including normal and abnormal ookinete in morphology) per 100 female gametocytes was calculated as the ookinete conversion rate.

### Parasite genetic cross

ICR mice were treated intraperitoneally with phenylhydrazine for gametocyte induction. Three days post-treatment, an equal number ($3 \times 10^6$) of asexual stage parasites from two different gene knockout lines were mixed and injected via the tail vein into the phenylhydrazine pre-treated mice. After 3 days, mouse blood with mixed gametocytes from two different parasite lines was collected from the mice and subjected for the in vitro gametocyte-gamete-zygote-ookinete development analysis using the in vitro ookinete culture described above.

### Mosquito transmission of the parasite

Approximately 100 female *Anopheles stephensi* mosquitoes were allowed to feed on an anesthetized mouse with 4–6% gametocytaemia for 30 min. To evaluate midgut infection of parasite, mosquito guts (*n* ~ 30) were dissected and stained with 0.1% mercurochrome 7 days post-feeding, and oocysts were tallied under a microscope. For quantifying salivary gland sporozoites, mosquito salivary glands (*n* ~ 30) were dissected 14 days after feeding, with the sporozoites counted using a hemocytometer. Transmission efficacy was assessed by allowing ~30 infected mosquitoes to feed on a naïve mouse for 30 min at day 14 post-feeding. Parasite transmission from mosquito to mouse was monitored 5 days later via Giemsa-stained thin blood films. These procedures were performed in triplicate.

### Flow cytometry analysis and sorting of male and female gametocytes

To analyze DNA content of male gametocytes, parasites containing gametocytes from the *DFsc7* or *DFsc7*;Δ*Rbpm1* lines were divided into two equal parts. One part was promptly fixed with 4% paraformaldehyde in PBS, while the other was exposed to exflagellation medium at 22 °C for 8 min to initiate gametogenesis before fixation. After staining with 4 μM Hoechst 33342 (Thermo Fisher Scientific, cat# 62249) for 10 min at room temperature and subsequent PBS washes, the samples were analyzed via flow cytometry on a BD LSRFortessa device (BD Biosciences, San Jose, CA, USA). Based on cell size and granularity, forward and side scatter signals were used to distinguish red blood cells from debris, doublets and white blood cells. Male gametocytes were identified by GFP fluorescence and analyzed for Hoechst 33342

fluorescence. For sorting gametocytes, parasites containing gametocytes were kept in GMB at 4 °C and sorted on a BD FACS AriaIII based on GFP and mCherry fluorescence for male and female gametocytes, respectively. Sorted gametocyte purity was verified by re-analysis of a sample fraction.

### Bulk RNA sequencing (RNA-seq)

Total RNA from $2 \times 10^7$ purified gametocytes was isolated using TRIzol (Thermo Fisher Scientific, cat#15596026) according to the manufacturer's instructions. RNA integrity was confirmed with an Agilent 2100 Bioanalyzer (Agilent Technologies, Palo Alto, CA, USA). mRNA was isolated with Oligo (dT) beads, fragmented, and reverse-transcribed to cDNA using random primers. Using DNA polymerase I, RNase H, dNTPs, and buffer, a second cDNA strand was synthesized. The resulting cDNA fragments were purified with the QIAQuick PCR Purification Kit (Qiagen, cat#28104), end-repaired, A-tailed, and ligated to Illumina sequencing adapters. The ligation products were size-selected using agarose gel electrophoresis, PCR amplified, and sequenced using the Illumina NovaSeq 6000 by Genedenovo Biotechnology Co., Ltd (Guangzhou, China).

### Differential expression analysis of RNA-seq data

Illumina-generated paired-end FASTQ files were trimmed using Trim Galore (v0.6.10)[75] (trim_galore --illumina -q 20 --paired --stringency 3 --length 25 -e 0.1 --fastqc --gzip) to remove the sequencing adapters and low quality reads. To refine the dataset, rRNA and tRNA were removed via a genome alignment program HISAT2 (v2.2.1)[76] (hisat2 -p 12 -q --unconc-gz). The cleaned reads, around 40 million per sample, were aligned to the *Plasmodium yoelii* 17X reference genome (PlasmoDB-62 release) using HISAT2 (hisat2 -p 12 -q). The resulting BAM files were sorted by position and indexed with SAMtools (v1.16.1)[77] (samtools -sort | samtools index). Mapped reads were summarized using featureCounts (v2.0.3)[78]. Gene expression analysis were performed in R (v4.2.1). Gene expression levels were normalized using transcripts per million (TPM) with the R package t-arae/ngscmdr (v0.1.0.181203)[79]. Differential expressed genes (DEGs, fold change > 2, and false discovery rate < 0.05) were assessed by the R package edgeR (v3.40.2)[80]. The volcano plot of DEGs were generated by the ggplot2 (v3.4.2)[81].

### Bioinformatic analysis of global intron retention

A GFF file containing genomic intron information was crafted using a perl script from agat package[82] and an in-house bash script, then converted into a BED file with BEDOPS convert2bed (v2.4.41)[83]. DeepTools bamCoverage (v3.5.1)[84] was used to generate the bigWig files for peak visualization in Integrative Genomics Viewer (IGV, v2.16.1)[38], and calculate the peak score of each exon and intron regions. Low expressed genes (TPM below 30) were excluded. To exclude potential false hits, introns with peak scores exceeding 50% of adjacent exons were discarded in parental parasites. In mutant parasites, introns with peak scores under 50% of neighboring exons were also omitted. Before differential intron retention analysis, the introns were normalized based on the gene expression level:

$$\text{Normalized intron counts} = \frac{\text{Intron region counts} \times 1000}{\text{Corresponding gene counts}}$$

Differentially retained introns (fold change > 2 and false discovery rate <0.05) were assessed with the R package edgeR (v3.40.2)[80]. These introns were further validated by IGV visualization and RT-PCR.

### Bioinformatic analysis of RBP in the *P. falciparum* and *P. berghei*

189 putative RBPs had been predicted in silico in the *P. falciparum*[32]. Among them, 179 RBPs have homologous proteins in *P. yoelii* and *P. berghei*. Differential expression analysis of the 179 RBPs between male and female gametocytes of *P. berghei* was based on the public dataset

by Yeoh, L.M., 2017[31]. The RNA-seq FASTQ files from NCBI SRA database (Accession: PRJNA374918) were processed using Trim Galore (v0.6.10) and HISAT2 (v2.2.1) for quality trimming and rRNA/tRNA removal, respectively. The cleaned reads were mapped to the *P. berghei* ANKA strain genome (PlasmoDB-62 release) using HISAT2 (v2.2.1), and the resulting BAM files were sorted and indexed with SAMtools (v1.16.1). Mapped reads were summarized using featureCounts (v2.0.3), and the differential expression analysis of RBPs was performed by the R package edgeR (v3.40.2). Differential expression analysis of the 189 RBPs between male and female gametocytes of *P. falciparum* is based on the public dataset from Lasonder E. 2016[30]. The RNA-seq FASTQ files from NCBI SRA (Accession: PRJNA305391) were processed similarly as above. The cleaned reads were mapped to the *Plasmodium falciparum* 3D7 reference genome. Given the absence of biological replicates in this dataset, differential expression analysis was performed by Cufflinks (v2.2.1)[85] (cuffdiff -p 8 --dispersion-method blind --library-norm-method geometric --library-type ff-firststrand). RBPs with fold change > 2 and false discovery rate <0.05 were considered differentially expressed. The volcano plot of differentially expressed RBPs were generated by the ggplot2 (v3.4.2).

## Antibodies and antiserum

The following primary antibodies were utilized: rabbit anti-HA (Cell Signaling Technology, cat#3724 S; IFA, 1:1000 dilution; IB, 1:1000 dilution), rabbit anti-mCherry (Abcam, cat# ab167453; IFA, 1:1000 dilution), rabbit anti-histone H3 antibody (Abcam, cat#ab1791; IFA, 1:1000 dilution), rabbit anti-Myc (Cell Signaling Technology, cat#2272 S; IFA, 1:1000 dilution; IB, 1:1000 dilution), mouse anti-α-Tubulin (Sigma-Aldrich, cat#T6199; IFA, 1:1000 dilution; IB, 1:1000 dilution; U-ExM, 1:500 dilution), mouse anti-β-Tubulin (Sigma-Aldrich, cat#T5201; IB, 1:1000 dilution) and mouse anti-HA (Santa Cruz Biotechnology, cat#sc-57592; IFA, 1:200 dilution). The secondary antibodies included: Alexa Fluor 555 goat anti-rabbit IgG (Thermo Fisher Scientific, cat#A-21428; IFA, 1:1000 dilution), Alexa Fluor 488 goat anti-rabbit IgG (Thermo Fisher Scientific, cat#A-31566; IFA, 1:1000 dilution), Alexa Fluor 555 goat anti-mouse IgG (Thermo Fisher Scientific, cat# A-21422; IFA, 1:1000 dilution; U-ExM, 1:500 dilution), Alexa Fluor 488 goat anti-mouse IgG (Thermo Fisher Scientific, cat#A-11001; IFA, 1:1000 dilution), Alexa Fluor 488 goat anti-mouse TER-119 (BioLegend, cat#116215; IFA, 1:500 dilution), Alexa Fluor 488 conjugated streptavidin (Invitrogen, cat# S32354; IFA, 1:1000 dilution), HRP-conjugated goat anti-rabbit IgG (Abcam, cat#ab6721; IB, 1:5000 dilution) and HRP-conjugated goat anti-mouse IgG (Abcam, cat#ab6789; IB, 1:5000 dilution). The antiserum, including rabbit anti-BiP (IB, 1:1000 dilution) and rabbit anti-P28 (IFA, 1:1000), were previously in-house prepared in the laboratory[86].

## Immunofluorescence assay

Parasites fixed in 4% paraformaldehyde in PBS were placed on poly-L-lysine-coated coverslips in a 24-well plate and centrifuged at 550 g for 5 min. They were then permeabilized with 0.1% Triton X-100 in PBS for 10 min at room temperature, blocked with 5% BSA/PBS at 4 °C overnight, and incubated with primary antibodies in 5% BSA/PBS for 1 h at room temperature. After three PBS washes, the samples were incubated with fluorescently labeled secondary antibodies in 5% BSA/PBS for 1 h at room temperature. Hoechst 33342 at a 1:5000 dilution in PBS was applied for 15 min at room temperature. Finally, the coverslips were washed, mounted in 90% glycerol, and sealed with nail varnish. Imaging was performed with a Zeiss LSM 780 confocal microscope at 100 × magnification.

## Ultrastructure expansion microscopy (U-ExM)

According to the method described in[87], gametocytes were fixed in 4% paraformaldehyde in PBS, then transferred to poly-D-lysine-coated coverslips in a 24-well plate and centrifuged. They were incubated in a 1.4% formaldehyde (Sigma-Aldrich, cat#F8775) and 2% acrylamide (Sigma-Aldrich, cat# A4058) mixture in PBS overnight at 37 °C. Afterward, the coverslips were gelled in a monomer solution containing 23% sodium acrylate (Sigma-Aldrich, cat#408220), 10% acrylamide, and 0.1% N,N'-methylenbisacrylamide (Sigma-Aldrich, cat#M1533) in PBS with tetramethylethylenediamine (TEMED) and ammonium persulfate (APS) at 37 °C for 1 h of polymerization. After polymerization, the coverslips were moved to a 6-well plate with denaturation buffer (200 mM SDS, 200 mM NaCl, 50 mM Tris-HCl, and pH 8.8) for 15 min at room temperature to detach the gels. The gels were denatured in 1.5 ml Eppendorf tubes with denaturation buffer at 95 °C for 30 min, incubated with ddH$_2$O at room temperature overnight in a 10 cm dish for the first round of expansion. The expanded gels were incubated with mouse anti-α-Tubulin antibody diluted in 2% BSA/PBS at room temperature for 3 h, washed 3 times with PBS, and incubated with anti-mouse Alexa 555 diluted in 2% BSA/PBS at room temperature for 3 h. After 3 washes in PBS, the gels were transferred into 10 cm dishes and incubated with ddH$_2$O at room temperature for the second round of expansion. Subsequently, gel blocks of ~5 mm × 5 mm were excised from the expanded gels and placed in the cavity well of cavity well microscope slides, covered with a coverslip, and imaged using a Zeiss LSM 980 confocal microscope.

## Protein extraction and immunoblot

Asexual blood parasites, gametocytes or ookinetes were lysed in RIPA buffer (Solaribio, cat#R0010) containing a protease inhibitor cocktail (MedChemExpress, cat#HY-K0010). After ultrasonication, the lysate was centrifuged at 14,000 g at 4 °C for 10 min. The resulting supernatant was mixed with SDS-PAGE loading buffer and heated at 95 °C for 5 min. Following SDS-PAGE separation, samples were transferred to a PVDF membrane (Millipore, cat#IPVH00010) and blocked with 5% milk in 1 × TBST (20 mM Tris-HCl pH 7.5, 150 mM NaCl, 0.1% Tween20) at 4 °C overnight. PVDF membranes were then incubated with primary antibodies at room temperature for 1 h. After washing with 1 × TBST, the membranes were incubated with an HRP-conjugated secondary antibody and then washed again with 1 × TBST. Finally, the membranes were visualized using a high-sensitivity ECL chemiluminescence detection kit (Vazyme, cat#E412-01), and the light emission was recorded either by X-ray film or by Azure Biosystems C280 (Azure Biosystems, USA).

## Isolation of nuclear and cytoplasmic fractions

The procedures were performed with modifications according to the previous study[88]. Nycodenz-purified gametocytes were first released from red blood cells by incubating them with 0.15% saponin/PBS on ice for 5 min and then washed twice with ice-cold PBS. The parasite pellet was resuspended in ice-cold lysis buffer (20 mM HEPES pH 7.9, 10 mM KCl, 1.5 mM MgCl$_2$, 1 mM EDTA, 1 mM EGTA, 1 mM DTT, and 0.65% Nonidet P-40) supplemented with protease inhibitor cocktail. The lysate was transferred to a 1 ml Dounce tissue grinder and homogenized gently for 80 strokes on ice. Nuclei were pelleted at 9000 g at 4 °C for 10 min, and the resulting supernatant represented cytoplasmic fractions. The nuclear pellet was washed twice with ice-cold lysis buffer before resuspension in one pellet volume of high salt buffer (20 Mm HEPES pH 7.8, 1 M KCl, 1 mM EDTA, 1 mM EGTA, and 1 mM DTT) supplemented with a protease inhibitor cocktail. After vigorous shaking at 4 °C for 30 min, the extract was centrifuged at 14,000 g at 4 °C for 10 min, and the resulting supernatant represented nuclear fractions. Immunoblotting was performed to analyze the proteins in each fraction.

## Protein immunoprecipitation

Nycodenz-purified gametocytes containing 3 × 10$^7$ male gametocytes were lysed in 1 ml lysis buffer (0.01% SDS, 20 mM Tris-HCl pH 8.0, 50 mM NaCl, 1 mM DTT) supplemented with protease inhibitor

cocktail. The lysate was transferred to a 1 ml Dounce tissue grinder and homogenized gently for 100 strokes on ice. The homogenate was transferred to an Eppendorf tube and incubated on ice for 10 min before centrifugation at 14,000 g at 4 °C for 10 min. The resulting supernatant was divided into two equal portions, with one portion mixed with 20 μl pre-balanced anti-GFP nanobody agarose beads (KT HEALTH, cat#KTSM1301) and the other portion mixed with anti-Myc nanobody agarose beads (KT HEALTH, cat#KTSM1306). Both portions were incubated at 4 °C for 2 h with rotation. The beads were then washed three times with lysis buffer before elution with SDS-PAGE loading buffer, followed by incubation at 95 °C for 5 min. Immunoblotting was performed on equal volumes of the supernatant samples.

### Transmission electron microscopy
Nycodenz-purified gametocytes were fixed at 8 mpa and 15 mpa in 2.5% glutaraldehyde in 0.1 M phosphate buffer at 4 °C overnight, as previously described[89]. Then, the samples were post-fixed in 1% osmium tetroxide at 4 °C for 2 h, treated *en bloc* with uranyl acetate, dehydrated, and embedded in Spurr's resin. Thin sections were sliced, stained with uranyl acetate and lead citrate, and examined in an HT-7800 electron microscope (Hitachi, Japan).

### TurboID-based proximity-labeling and biotinylated protein pull-down
Nycodenz-purified gametocytes containing $1 \times 10^8$ male gametocytes from either the *Rbpm1::TurboID* or *Rbpm1::T2A::TurboID* line were incubated with 50 μM biotin (Sigma-Aldrich, cat#B4639) at 37 °C for 20 min. After biotinylation, the parasites were pelleted, washed thrice with 1 ml ice-cold PBS to remove excess biotin, and then lysed with RIPA buffer containing a protease inhibitor cocktail via ultrasonication. The lysate was incubated on ice for 10 min before centrifugation at 14,000 g at 4 °C for 10 min. The supernatant was then mixed with 50 μl pre-balanced streptavidin sepharose (Thermal Scientific, cat#SA10004) at 4 °C overnight. The beads were washed five times with 1 ml ice-cold RIPA buffer and then washed five times with 1 ml ice-cold PBS. The washed beads were resuspended in 200 μl 100 mM Tris-HCl pH 8.5 followed by digestion with 1 μg trypsin at 37 °C overnight.

### Peptide desalting and mass spectrometry
Trifluoroacetic acid (TFA; Sigma-Aldrich, cat#T6508) was added to the trypsin-digested sample to a final concentration of 1%, and the precipitation of sodium deoxycholate was removed by centrifugation. The resulting supernatant was desalted using in-house-made StageTips that were packed with SDB-RPS (3 M EMPORE, cat#2241) and conditioned with 50 μl of 100% acetonitrile (ACN; Sigma-Aldrich, cat# 34851). After loading the supernatant onto the StageTips, centrifugation was performed at 3000 g for 5 min. The StageTips were then washed twice with 50 μl of 1% TFA/isopropyl alcohol (Sigma-Aldrich, cat# I9030) followed by a wash with 50 μl of 0.2% TFA. The peptides were eluted in glass vials (CNW Technologies, cat# A3511040) using 80% ACN/5% NH$_4$OH and dried at 45 °C using a vacuum centrifuge (Eppendorf, Hamburg, Germany, cat#5305). The peptide samples were resolved in 2% ACN/0.1FA for LC-MS analysis. Liquid chromatography was performed on a high-pressure nano-flow chromatography system (Elute UHPLC, Bruker Daltonics). Peptides were separated on a reversed-phase column (40 cm × 75 μm i.d.) at 50 °C packed with 1.8 μm 120 Å C18 material (Welch, Shanghai, China) with a pulled emitter tip. A solution is 0.1% FA in H$_2$O, and B solution is 0.1% FA in ACN. The gradient time is 60 min and the total run time is 75 min including washes and equilibration. Peptides were separated with a linear gradient from 0 to 5% B within 5 min, followed by an increase to 30% B within 55 min and further to 35% B within 5 min, followed by a washing step at 95% B and re-equilibration. LC was coupled online to a hybrid TIMS quadrupole time-of-flight mass spectrometer (Bruker timsTOF Pro) via a CaptiveSpray nano-electrospray ion source. We performed data-dependent data acquisition in PASEF mode with 10 PASEF scans per topN acquisition cycle. Singly charged precursors were excluded by their position in the m/z-ion mobility plane and precursors that reached a 'target value' of 20,000 a.u. were dynamically excluded for 0.4 min. We used 100 ms to accumulate and elute ions in the TIMS tunnel. The MS1 m/z-range was acquired from 100 to 1700, and the ion mobility range from 1.5 to 0.7 V cm$^{-2}$. For data-independent acquisition, we adopted the isolation scheme of 25 Da × 32 windows to cover 400-1200 mz. DIA files (raw) files were input to DIA-NN (v1.8.1)[90] FASTA files downloaded from https://www.uniprot.org (UP000072874) were added. "FASTA digest for library-free search" and "Deep learning-based spectra, RTs, and IMs prediction" were enabled. "Generate spectral library" was also enabled. "Protein inference" was set to "gene". Other parameters were kept at their default settings. The protein groups and precursor lists were filtered at 1% FDR, using global q-values for protein groups and both global and run-specific q-values for precursors.

### RNA isolation, RT-PCR and RT-qPCR
Total RNA was extracted from parasites using TRIzol reagent. cDNA was synthesized with the HiScript II 1st Strand cDNA Synthesis Kit (Vazyme, cat#R212-02), using provided random hexamers, and utilized for PCR or qPCR analysis. qPCR was performed using 2×RealStar Green Fast Mixture (GenStar, cat#A301-101) with the following cycling program: a single incubation at 95 °C for 30 s, followed by 40 cycles (95 °C for 5 s, 60 °C for 40 s) on a CFX96 Real-Time PCR System (Bio-Rad, Hercules, CA, USA). The housekeeping gene *gapdh* (PY17X_1330200) was used as a reference gene in the RT-qPCR. The relative expression was calculated using the $2^{-\Delta\Delta Ct}$ method. The primers used for RT-PCRs and RT-qPCRs are listed in Supplementary Table 2.

### UV crosslinking RNA immunoprecipitation (UV-RIP)
The Nycodenz-purified gametocytes, containing $6 \times 10^7$ male gametocytes in 6 ml ice-cold PBS, were placed in 10 cm dishes. Subsequently, they were irradiated using an HL-2000 HybriLinker (UVP, Upland, CA, USA) with 254 nm UV light at intensities of 400 mJ/cm$^2$ and 200 mJ/cm$^2$. The gametocytes were then collected, centrifuged, and resuspended in 1 ml lysis buffer (1% TritonX-100, 50 mM Tris-HCl pH 7.4, 150 mM NaCl, 1 mM EDTA, 1 mM EGTA) supplemented with 400 U/ml RNaseOUT (Thermo Fisher Scientific, cat#10777019) and a protease inhibitor cocktail. The lysate was transferred to a 1 ml Dounce tissue grinder and gently homogenized for 100 strokes on ice. The homogenate was then transferred to a tube and incubated at 4 °C for 25 min with rotation, followed by treatment with 30 U TURBO DNase (Thermo Fisher Scientific, cat#AM2238) at 37 °C for 15 min. The lysates were centrifuged at 14,000 g and 4 °C for 10 min. The supernatant was divided into two equal parts. One part was mixed with 20 μl of anti-GFP nanobody agarose beads (KT HEALTH, cat#KTSM1301), and the other part was mixed with 20 μl of anti-HA nanobody agarose beads (KT HEALTH, cat#KTSM1305). The mixtures were incubated with rotation at 4 °C for 2 h. The beads were washed six times with 500 μl RIP wash buffer (Millipore, cat#CS203177) at 4 °C and then incubated with 117 μl RIP wash buffer, 15 μl 10% SDS and 18 μl 10 mg/ml proteinase K (Millipore, cat#CS203218) at 55 °C for 30 min. RNA was isolated using phenol-chloroform extraction, and the purified RNA was reverse transcribed with random hexamer primers and determined by RT-qPCR.

### In vitro RNA transcription (IVT)
To prepare biotinylated probes for Fig. 6O, P, IVT templates with T7 RNA polymerase promoter were obtained by PCR using the *P. yoelii* genome as a template. For Fig. 7H, IVT templates with T7 RNA polymerase promoter were obtained by PCR using the plasmid used in the *bfp* reporter assay as a template. Supplementary Table 2 provides a list of primers used to obtain the IVT templates. Subsequently,

Biotinylated RNA was produced using a MEGAscript kit (Thermo Fisher Scientific, cat#AM1334) and a biotin RNA labeling mix (Roche, cat#11685597910). To create a 20 μl reaction volume, 1 μg of PCR-amplified IVT templates were incubated at 37 °C for 2 h with 2 μl of 10× reaction buffer, 2 μl of T7 RNA polymerase enzyme mix, 2 μl of biotin RNA labeling mix, and RNase-free water. The DNA templates were then removed from the RNA using TURBO DNase, and the biotinylated RNA was purified using the RNAclean Kit (TIANGEN, cat#4992728). In this process, the *kinesin8b* I4 probe, *kinesin8b* I1 probe, *PF16* E1 probe, and *PF16* I1 probe all have a length of 500 nt. Additionally, the *kinesin8b* I4 probe, *kinesin8b* I1 probe, and *PF16* I1 probe span the corresponding intron sequences.

### RNA pull-down
Biotinylated RNA pull-down was performed using an RNA pull-down Kit (BersinBio, cat# Bes5102) following the manufacturer's protocol. Briefly, 1 μg of biotinylated RNA was denatured at 90 °C for 2 min and immediately cooled on ice for 2 min. The denatured RNA was then incubated with RNA structure buffer and RNase-free water at room temperature for 20 mi to facilitate RNA secondary structure formation. For cell lysate preparation, Nycodenz-purified gametocytes containing $3 \times 10^7$ male gametocytes were lysed by RIP buffer, and the resulting lysate was centrifuged at 14,000 $g$ at 4 °C for 10 min. The supernatant was then incubated with DNase I and agarose beads to remove the chromosomes, followed by incubation with folded RNAs, streptavidin-coupled beads, and RNase inhibitor at room temperature for 2 h. The beads were subsequently washed five times with NT2 buffer at 4 °C, and proteins were retrieved from the beads by rinsing them with protein elution buffer. The retrieved proteins were then subjected to immunoblot assay.

### *bfp* reporter assay
The Nycodenz-purified gametocytes from either *DFsc7* or *DFsc7;ΔRbpm1* lines, which contain a *bfp* expression cassette in the *p230p* locus, were suspended in 200 μl of GMB. The samples were then transferred to a 15 mm glass bottom cell culture dish and imaged using a Zeiss LSM 780 confocal microscope at room temperature with 100× magnification. The laser illumination was set at 561 nm (mCherry), 491 nm (GFP), and 405 nm (BFP). BFP-positive parasites indicated that the intron in the *bfp* expression cassette had been spliced.

### Other bioinformatic analysis and tools
The genomic sequences of target genes were downloaded from the PlasmoDB database (http://plasmodb.org/plasmo/). The sgRNAs of target gene were designed using EuPaGDT (http://grna.ctegd.uga.edu/). The analysis of flow cytometry data was performed using the FlowJo software (Tree Star, Ashland, OR, USA). The Gene Ontology (GO) enrichment analysis was performed using PlasmoDB. Statistical analysis was performed using GraphPad Prism (GraphPad Software Inc., San Diego, CA, USA) with either a two-tailed Student's *t*-test or Mann-Whitney test as appropriate. Error bars represent the standard error of the mean (SEM) for triplicate experiments. *p* values were indicated in the figures above the two groups being compared, with a value <0.05 considered significant. The protein signal on the blotting membrane was quantified using ImageJ software (NIH, Bethesda, MD, USA), and the background was subtracted from each signal. Each signal was then normalized to the Bip signal.

### Reporting summary
Further information on research design is available in the Nature Portfolio Reporting Summary linked to this article.

## Data availability
All relevant data in this study are submitted as supplementary source files. Source data are provided with this paper. The RNA-seq data for the *P. yoelii* male- and female gametocyte transcriptome have been deposited in the Gene Expression Omnibus database under the accession number GSE222860. The RNA-seq data for male gametocyte transcriptome of the *P. yoelii Rbpm1* knockout parasite line is available under accession number GSE223170. The mass spectrometry proteomic data have been deposited in the ProteomeXchange with identifier PXD044094 (https://www.iprox.cn//page/project.html?id=IPX0006804000). Source data are provided with this paper.

## Code availability
All code and supporting files for transcriptome and intron retention analysis in this study were available in Zenodo (https://doi.org/10.5281/zenodo.10979262).

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

## Acknowledgements

This work was supported by the National Natural Science Foundation of China (32170427 by J.Y., 32270503 by H.C.), the Natural Science Foundation of Fujian Province (2021J01028 by J.Y.), and the 111 Project sponsored by the State Bureau of Foreign Experts and Ministry of Education of China (BP2018017 by J.Y.).

## Author contributions

J.G. and J.Y. designed the study. J.G., P.W., X.Z.(Xiaoming Zhang), W.L., and X.Z. (Xiaolong Zhang) generated the modified parasites. J.G. performed phenotype analysis, protein analysis, electron microscopy, RNA analysis, and reporter assays. P.W. conducted mass spectrometry and protein analysis. X.M. performed the bioinformatics analysis. L.J., J.L., H.C., and J.Y. supervised the work. J.G., X.M., and J.Y. wrote the manuscript.

## Competing interests

The authors declare no competing interests.
