## [Peer Review File · Nature Communications]

An axonemal intron splicing program sustains Plasmodium male developmentREVIEWER COMMENTS

Reviewer #1 (Remarks to the Author):

The Plasmodium parasite has a complex lifecycle, which includes transition from the asexually replicating blood stage to a sexual stage inside the vertebrate host. This stage includes the development of male and female gametocytes, which subsequently further differentiate to gametes, followed by gamete fusion, leading to zygote formation and continued development inside the midgut of the mosquito vector. There are still major knowledge gaps linked to the regulation of sexual cell fate and general development of male and female gametocytes. The study by Guan et al. focus on the rodent infectious *P. yoelii* species where they have identified an RNA-binding protein (RBP), which they go through lengths to show that it is essential specifically for male gametogenesis. The study includes localization assays of the target protein, a functional knockout of "RBPm1" followed by phenotype characterization and analysis of downstream effects, including the identification of specific intron splicing. The study is highly ambitious and provides information that is important to the field.

However, since the study is specifically focused on the expression, regulation and target effects of one protein, it would significantly increase the impact if there were cross-validations in the human-infectious *P. falciparum* species, which can be i) easily cultured and ii) exflagellated in vitro or used for mosquito infections. Several orthologous genes among the Plasmodium species have shown alternative functions, and it has been demonstrated that gametocytogenesis is significantly different in *P. falciparum* compared to other Plasmodium species.

A complete repeat of this ambitious study in *P. falciparum* is not necessary, but the addition of cross-validation of key findings is in my mind necessary to increase the impact. Further, since *P. falciparum* has a significantly longer gametocyte developmental stage, it would be interesting to see the timing of expression of the PY17X_0716700 ortholog and potentially other male-specific RBPs as well in *P. falciparum*.

In addition, the manuscript would highly benefit from going through a thorough language check throughout the text, prior to a potential revision.

Reviewer #2 (Remarks to the Author):

This manuscript elucidates the mechanism of a novel RNA-binding protein, RBPm1, in regulating the expression of axonemal genes in Plasmodium male gametes. The authors present compelling evidence demonstrating RBPm1's binding to the introns of mRNAs from axonemal genes, orchestrating their splicing by facilitating the assembly of complex E. The study features an impressive and comprehensive collection of transgenic parasites, underscored by a meticulously designed experimental approach. The manuscript is well-crafted and easily comprehensible. The topic holds substantial relevance for the malaria and parasitology community, shedding light on the molecular mechanisms through which this novel protein governs the expression of genes crucial for male gamete fertility. This understanding could pave the way for innovative targets in developing interventions to block Plasmodium transmission.

I note a few points for manuscript improvement.

The absence of additional analyses on the transcriptome data from male and female gametes is surprising. While acknowledging the manuscript's primary focus on RBPs, supplementing information such as sequencing quality, number of genes, heat maps for differential expression, and gene ontology in the supplementary materials would enhance completeness. Additionally, clarification on whether the sequencing data has been deposited in a database system, along with the corresponding access link, is essential.

The authors could not define an RBPm1 binding motif in the target introns using bioinformatics. An alternative approach is to perform mutagenesis in target and non-target introns to try to identify nucleotides important for the binding using RNA pull-down assays.

In the model proposed in Figure S11M and the discussion, the authors suggest that the weak transcription of axonemal genes together with IR prevents their protein expression in females. Could there be other explanations for the low transcript levels, like the possibility that there is a similar level of transcription as in males but a faster mRNA decay due to RI?

Minor comments:

40: the order should be "escape from host erythrocytes and develop into fertile gametes". The gametes are not fertile until after egress from the RBC.

81: Substitute "was" with "is." "Was" implies a past state that no longer exists.

107: wild type instead of wildtype

110: substitute at for in

135: I could not find any data showing that disruption of Rbpm1 does not affect asexual blood stage proliferation. The data shows that there is no effect on gametocyte formation.

351-352: It is not clear why the "expected" binding to the "target" intron and not the "neighboring" introns. This could be better clarified for the non-expert reader.

404: "significantly more", could the authors quantify the band intensity and calculate the ratio against the bands in the input lane?

458: consider editing to "...entering the mosquito midgut..."

Figure 2D, G, H, I and L: please verify the statistical analysis. T-test and Mann-Whitney are not appropriate. One-way ANOVA with multiple comparison should be used.

Figure 3B: "The numbers indicate the relative intensities...", relative intensity to what? BiP? Please clarify.

Figure 3D: Consider editing the second sentence to "Inset panels show longitudinal sections..."

Reviewer #3 (Remarks to the Author):

The study "An axonemal intron splicing program sustains Plasmodium male development" by Guan and colleagues describes the function of the RNA-binding protein Rbpm1 during male gametogenesis of the rodent malaria pathogen *Plasmodium yoelii*. The study is very attractive and uses a wide range of state-of-the-art methods, including mutant generation and omics analysis. However, I see several conceptual weaknesses in the study that would call the results into question. These are my comments:

The study is based on the statement that Rbpm1 is expressed in male gametocytes and is important for exflagellation. The authors use immunostaining of alpha and beta tubulin to label male gametocytes. However, the two tubulin proteins are not specific for male gametocytes, but are components of the microtubule cytoskeleton, as it is found in all parasite stages. Microtubules are expressed in female and male gametocytes, including below the plasma membrane. In the study, the authors provide no evidence of male expression of tubulins, nor do they refer to published studies. All subsequent experiments on gender specificity based on this statement must then be questioned.

During their analyses, the authors silence a number of genes and examine their exflagellation behaviour, such as the genes encoding kinesin 8b and PF16. However, no analyses of intraerythrocytic growth or gametocytogenesis and female gametogenesis are shown. Of course, if growth rates or gametocytogenesis are already impaired, then this would also have an indirect impact on exflagellation, without the genes being involved in exflagellation at all.

In the course of the studies on Rbpm1, the authors carried out comparative transcriptome studies on Rbpm1-KO versus wild type. However, the authors rush through the experiment; no information is given on the number of deregulated genes, nor are their GeneIDs and GO terms given. Important genes may be affected that indicate a function of Rbpm1 outside of exflagellation, so the data need to be disclosed.

The authors performed BioID analyses to identify interaction partners of Rbpm1. 113 significant

proteins are identified, but their names or GeneIDs are not mentioned. Here too, the question arises as to whether there are interaction partners that indicate a function of Rbpm1 outside of exflagellation. Therefore, the data should also be disclosed here. String analysis would be helpful to visualize protein interaction clusters.

Further comments:

Female gametogenesis is measured and depicted in Figure 2F. However, only one image is shown and it remains unclear how many cells were evaluated.

Co-immunoprecipitations are shown in Figures 6E, F, G. However, it can be seen from the molecular weights and the original blots that the figures do not represent the same blots, but that the bands were put together from different blots. The data is therefore questionable and the negative and loading controls are inconclusive.

The authors report that all IR genes are male-specific (Fig. S4A). I do not understand, how they confirmed this.

The abstract and introduction contain many grammatical errors that, surprisingly, are not present in the rest of the text. The introduction is also very general. Generalized statements about the fact that RNA-binding proteins are hardly known in male gametocytes are also not true; some recent studies report such proteins (e.g. Hanhsen et al., 2022; Russell et al., 2023, Farrukh et al., 2023, etc).

Response to Reviewer Comments on the manuscript [NCOMMS- 23-04049]:

Reviewer#1

The Plasmodium parasite has a complex lifecycle, which includes transition from the asexually replicating blood stage to a sexual stage inside the vertebrate host. This stage includes the development of male and female gametocytes, which subsequently further differentiate to gametes, followed by gamete fusion, leading to zygote formation and continued development inside the midgut of the mosquito vector. There are still major knowledge gaps linked to the regulation of sexual cell fate and general development of male and female gametocytes. The study by Guan et al. focus on the rodent infectious *P. yoelii* species where they have identified an RNA-binding protein (RBP), which they go through lengths to show that it is essential specifically for male gametogenesis. The study includes localization assays of the target protein, a functional knockout of “RBPm1” followed by phenotype characterization and analysis of downstream effects, including the identification of specific intron splicing. The study is highly ambitious and provides information that is important to the field.

Response: We thank the reviewer for the nice comments on our work.

However, since the study is specifically focused on the expression, regulation and target effects of one protein, it would significantly increase the impact if there were cross-validations in the human-infectious *P. falciparum* species, which can be i) easily cultured and ii) exflagellated in vitro or used for mosquito infections.

Several orthologous genes among the Plasmodium species have shown alternative functions, and it has been demonstrated that gametocytogenesis is significantly different in *P. falciparum* compared to other Plasmodium species. A complete repeat of this ambitious study in *P. falciparum* is not necessary, but the addition of cross-validation of key findings is in my mind necessary to increase the impact.

Response: Thank the reviewer for the suggestion. Since established, my lab utilized the rodent malaria parasite *P. yoelii* to study malaria transmission biology. Due to the lack of a stable supply of human blood for the in vitro culture of *P. falciparum* as well as the lack of a biosafety-level facility for mosquito infection experiments of *P. falciparum*, there is no platform for the *P. falciparum* experiments in my lab and the university. Currently, we can't perform phenotype analysis and gene function of RBPm1 in the *P. falciparum*. In the manuscript, we concluded the function of RBPm1 in the *P. yoelii*, and in the discussion we only speculate the RBPm1's function of *P. falciparum*.

To further address the concern of Reviewer#1 and strengthen the cross-validations of RBPm1 in other *Plasmodium* species, we re-analyzed the public single-cell transcriptomic RNA-seq data of *P. berghei* WT and the *md5* (*rbpm1* ortholog) knockout parasites (raw data: ENA: PRJEB44892, <https://zenodo.org/records/7317469>) provided by Russell *et al.* (PMID: 36634679, 2023). We found evidence supporting the conserved function of RBPm1 in regulating the axonemal intron splicing in *P. berghei* (see the picture in the next pages, not included in the revised manuscript).

A. Single-cell transcriptomics analysis and Uniform Manifold Approximation and Projection (UMAP) plot of 2599 parasitized red blood cells from *P. berghei* WT and nine mutant parasites, including the $\Delta md5$ (*rbpm1* ortholog) mutant, as adapted from Russell *et al* 2023. Cells were colored by genotype. Asexual blood stages (ABS), male gametocytes, and female gametocytes were separated in the UMAP plots.

B. Validation of the UMAP plot in **A** based on the expression of six known stage-specific genes.

C. ABS and male gametocytes of WT and $\Delta md5$ were selected from **A**.

D. Clustering of parasite population based on global intron splicing levels.

In the work by Russell *et al.*, high-resolution single-cell transcriptomics were performed for ten mutant parasites of *P. berghei*, including WT and the $\Delta md5$ mutant. Re-analysis of the transcriptomic datasets found different patterns for global intron splicing in male gametocytes between WT and $\Delta md5$ (Figure D). Meanwhile, the pattern in the asexual blood stages (ABS) seemed not different between WT and $\Delta md5$ (Figure D).

F 27 genes with intron retention

$\Delta md5$ male	WT male	$\Delta md5$ ABS	WT ABS	P. berghei intron identity	Orthologous genes in P. yoelii	Product Description
●	●	●	●	PBANKA_1029400 I5	PY17X_1031800	Conserved Plasmodium protein, unknown function
●	●	●	●	PBANKA_1447900 I1	md2 (PY17X_1450400)	Male development protein MD2
●	●	●	●	PBANKA_0827100 I1	NA	Protein kinase
●	●	●	●	PBANKA_0721100 I2	drc1 (PY17X_0721100)	Dynein regulatory complex protein
●	●	●	●	PBANKA_1029400 I1	PY17X_1031800	Conserved Plasmodium protein, unknown function
●	●	●	●	PBANKA_1029400 I2	PY17X_1031800	Conserved Plasmodium protein, unknown function
●	●	●	●	PBANKA_1329200 I1	dbc (PY17X_1333900)	Dynein beta chain
●	●	●	●	PBANKA_1402400 I3	PY17X_1404000	GINS complex subunit Psf3
●	●	●	●	PBANKA_1316500 I2	PY17X_1320300	Conserved Plasmodium protein, unknown function
●	●	●	●	PBANKA_1352100 I3	PY17X_1357300	SprT-like domain-containing protein
●	●	●	●	PBANKA_0504300 I2	PY17X_0505400	Flagellar outer arm dynein-associated protein
●	●	●	●	PBANKA_0212600 I1	PY17X_0214000	EF hand domain-containing protein
●	●	●	●	PBANKA_1308000 I9	PY17X_1311800	Conserved Plasmodium protein, unknown function
●	●	●	●	PBANKA_0520700 I16	PY17X_0521800	WD repeat-containing protein
●	●	●	●	PBANKA_0302200 I3	drc2 (PY17X_0302800)	Dynein light chain
●	●	●	●	PBANKA_0507800 I3	PY17X_0508900	Conserved protein, unknown function
●	●	●	●	PBANKA_1238300 I2	drc1 (PY17X_1241500)	Dynein light chain Tctex-type
●	●	●	●	PBANKA_1138600 I2	PY17X_1140000	Conserved protein, unknown function
●	●	●	●	PBANKA_1329100 I10	PY17X_1333800	Plasmeprin VIII
●	●	●	●	PBANKA_1242100 I1	PY17X_1245300	Methionine aminopeptidase 1a
●	●	●	●	PBANKA_0615700 I10	dhc6 (PY17X_0603800)	Dynein heavy chain
●	●	●	●	PBANKA_1320100 I1	PY17X_1323900	Conserved Plasmodium protein, unknown function
●	●	●	●	PBANKA_1018200 I2	PY17X_1019700	MORN repeat protein
●	●	●	●	PBANKA_0822000 I4	PY17X_0825300	Conserved Plasmodium protein, unknown function
●	●	●	●	PBANKA_1015400 I15	PY17X_1016900	CPW-WPC family protein
●	●	●	●	PBANKA_1236600 I5	PY17X_0301200	Conserved Plasmodium protein, unknown function
●	●	●	●	PBANKA_0614800 I1	PY17X_0617500	CRAL/TRIO domain-containing protein
●	●	●	●	PBANKA_1358200 I2	PY17X_1363900	Cytochrome c oxidase subunit ApiCOX16
●	●	●	●	PBANKA_1135500 I10	PY17X_1137000	High mobility group protein B4

Scaled average intron retention: -2 -1 0 1 2
 Cells with intron detected (%): 0.25 0.5 0.75 1

E. Global intron splicing analysis identified 27 intron-retained genes (light and dark blue dots) in male gametocytes of the $\Delta md5$ versus WT in the *P. berghei*. Among these, the orthologs of 12 genes (dark blue dot) also showed IR events in the *P. yoelii*.

F. Information of the identified 27 intron-retained genes in the gametocytes of the *P. berghei* $\Delta md5$. Those genes sharing orthologs with RBPm1's target genes in the *P. yoelii* were highlighted in dark blue, corresponding to **E**.

We further dug the intron retention (IR) events in the $\Delta md5$ male gametocytes and found 27 genes with IR after loss of *md5* compared to WT (Figure E). Notably, among these 27 genes, the orthologs of 12 genes (dark blue in letter) showed IR events in the *P. yoelii* validated in our study. These results strongly suggested the conserved function and mechanism of RBPm1 in the *P. berghei* as that in the *P. yoelii*.

Further, since *P. falciparum* has a significantly longer gametocyte developmental stage, it would be interesting to see the timing of expression of the PY17X_0716700 ortholog and potentially other male-specific RBPs as well in *P. falciparum*.
Response:

We went to the database of “Malaria Cell Atlas-*Plasmodium berghei* Atlas”, which integrated the single-cell RNA-seq data of *P. berghei* parasite provided by Russell *et al* (PMID:36634679, 2023). The website is <https://www.malariacellatlas.org/atlas/pb/>. Uniform Manifold Approximation and Projection (UMAP) visualization of single-cell transcriptomes from across intraerythrocytic asexual and sexual stages showed that the *Rbpml* ortholog (PBANKA_0716500) in the *P. berghei* was transcribed primarily in male gametocytes, and was expressed at a low level in developing gametocytes (see the picture in the next page, not included in the revised manuscript).

We also went to the database of “Malaria Cell Atlas-*Plasmodium falciparum* Atlas”, which integrated the single-cell RNA-seq data of *P. falciparum* parasite from two studies (Reid AJ *et al*. PMID: 29580379, 2018, and Real E *et al*. PMID: 34045457, 2021). The website is <https://www.malariacellatlas.org/atlas/plasmodium-falciparum-atlas/>. UMAP visualization of single-cell transcriptomes from across intraerythrocytic asexual and sexual stages showed that the *Rbpml* ortholog (PF3D7_0414500) in *P. falciparum* had dominant transcription in early and late male gametocytes (see the picture in the next page, not included in the revised manuscript).

Together, these data indicated that the *Rbpml* ortholog genes displayed a similarly male-specific expression in the *P. berghei* and *P. falciparum* parasites.

In addition, the manuscript would highly benefit from going through a thorough language check throughout the text, prior to a potential revision.
 Response: Thank the reviewer for pointing it out. We have a professional check the grammar and language throughout the manuscript.

Reviewer#2

This manuscript elucidates the mechanism of a novel RNA-binding protein, RBPm1, in regulating the expression of axonemal genes in Plasmodium male gametes. The authors present compelling evidence demonstrating RBPm1's binding to the introns of mRNAs from axonemal genes, orchestrating their splicing by facilitating the assembly of complex E. The study features an impressive and comprehensive collection of transgenic parasites, underscored by a meticulously designed experimental approach. The manuscript is well-crafted and easily comprehensible. The topic holds substantial relevance for the malaria and parasitology community, shedding light on the molecular mechanisms through which this novel protein governs the expression of genes crucial for male gamete fertility. This understanding could pave the way for innovative targets in developing interventions to block Plasmodium transmission.

Response: we thank the reviewer for the nice comments on our work.

I note a few points for manuscript improvement. The absence of additional analyses on the transcriptome data from male and female gametes is surprising. While acknowledging the manuscript's primary focus on RBPs, supplementing information such as sequencing quality, number of genes, heat maps for differential expression, and gene ontology in the supplementary materials would enhance completeness.

Response:

Thank the reviewer's suggestion. In the revised manuscript, we added information on the gametocyte transcriptome data (including the *DFsc7* female gametocyte, *DFsc7* male gametocyte, and *DFsc7;ΔRbpm1* male gametocytes). We analyzed the RNA-sequencing quality, number of genes, heat maps for differential expression, and gene ontology.

List of the relevant files added in the zipped source data file:

Source data Figure 1. RNA-seq information and analysis of the transcriptome data

Source data Table 1. List of differentially expressed genes between male and female gametocytes of the *P.yoelii* *DFsc7* line

Source data Table 2. List of differentially expressed genes in male gametocytes between *DFsc7* (Parental) and *DFsc7;ΔRbpm1* (Mutant) lines of the *P.yoelii*

Additionally, clarification on whether the sequencing data has been deposited in a database system, along with the corresponding access link, is essential.

Response:

We deposited the RNA-seq data and the proteomic data in the public database. In the manuscript, we had a statement of Data availability "RNA-seq data for the *P. yoelii* male- and female-specific gametocyte transcriptome is available via the Gene Expression Omnibus database under the accession number GSE222860 (secure token: yvgvagewhxitil)

<https://www.ncbi.nlm.nih.gov/geo/query/acc.cgi?acc=GSE222860>. RNA-seq data for male gametocyte transcriptome of the *P. yoelii Rbpm1* knockout parasite line is available under accession number GSE223170 (secure token: mxobemwslfolbov) <https://www.ncbi.nlm.nih.gov/geo/query/acc.cgi?acc=GSE223170>. The mass spectrometry proteomic data can be accessed through ProteomeXchange with identifier PXD044094 (<https://www.iprox.cn//page/project.html?id=IPX0006804000>)”.

The authors could not define an RBPm1 binding motif in the target introns using bioinformatics. An alternative approach is to perform mutagenesis in target and non-target introns to try to identify nucleotides important for the binding using RNA pull-down assays.

Response:

To define the RBPm1 binding or recognition motif in the target introns, we previously analyzed the 30 RBPm1-targeted introns. Using the motif search software MEME, a potential RNA motif “TA(G/T)GCA” was detected in 29 out of the 30 RBPm1-targeted introns (statistically significant, E-value= 2.8×10^{-7}). To test the importance of this motif, we mutated the motif sequence in three different target introns (*Kinesin8b* Intron1, *PF16* Intron1, and *dlc1* Intron4).

The WT and mutated introns were inserted into the *bfp* gene, and the intact *bfp* transcript driven by the *hsp70* 5'-UTR and the *dhfr* 3'-UTR was integrated into the *p230p* locus of the *DFsc7* parasite line using CRISPR-Cas9, generating a total of six transgenic lines for the *bfp* reporter assays.

The *bfp* reporter assays for intron splicing analysis showed that mutation of the TA(G/T)GCA motif in either *Kinesin8b* Intron1, *PF16* Intron1 or *dlc1* Intron4 had no significant effect on the BFP expression in male gametocytes. These results indicated that mutation of the TA(G/T)GCA motif had no effect on the recognition of these three introns by RBPm1 and that the motif of “TA(G/T)GCA” does not play an important role in the intron recognition by RBPm1 (see the picture in the next page, not included in the revised manuscript).

A. A potential RNA motif of TA(G/T)GCA detected from 30 RBPm1-target introns by MEME motif discovery algorithm.

B. Schematics showing the mutation in the motif of TA(G/T)GCA in three RBPm1-target introns *PF16* intron1, *dlc1* intron4, and *kinesin8b* intron1. One motif is in *PF16* intron1 and *dlc1* intron4, while two motifs in *kinesin8b* intron1. Red dots indicate the positions of the mutation (in red letters) within the introns.

C. The *bfp* reporter assays for intron splicing analysis showed that mutation of the TA(G/T)GCA motif in the *PF16* intron1 did not affect BFP expression in male gametocytes. Upper panel: A transgenic line *BFP-PF16 I1* with a wildtype *PF16* intron1 (*PF16 I1*, purple line) - inserted *bfp* cassette integrated at the *p230p* locus of the *DFsc7* line. BFP was specifically detected in male gametocytes. Lower panel: Insertion of *PF16 I1* with the TA(G/T)GCA motif mutation (*PF16 I1* mutant) did not affect BFP expression in male gametocytes. Scale bars: 5 μ m.

D. Analysis of the mutation in the motif of TA(G/T)GCA in the *dlc1* intron4 (*dlc1 I4*). Similar design as in C.

E. Analysis of the mutation in the motif of TA(G/T)GCA in the *kinesin8b* intron1 (*Kin8b I1*). Similar design as in C.

We further took the suggestion of Reviewer#2, and performed mutagenesis in the RBPm1 target intron *PF16* Intron1 (276 nt in length). We sought to identify the sub-region(s) critical for the RBPm1 recognition. Five partly overlapping fragments (F1: 3-71 nt, F2: 50-121 nt, F3: 109-179 nt, F4: 169-230 nt, and F5: 231-274 nt) were deleted and five sequences of *PF16* Intron1 with each fragment truncated (Δ F1, Δ F2, Δ F3, Δ F4, and Δ F5) were obtained (see the picture in the next page, not included in the revised manuscript).

The WT and mutated introns of *PF16* Intron1 were inserted into the *bfp* gene, and the intact *bfp* transcript driven by the *hsp70* 5'-UTR and the *dhfr* 3'-UTR was integrated into the *p230p* locus of the *DFsc7* parasite line using CRISPR-Cas9, generating a total of six transgenic lines for the *bfp* reporter assays.

The *bfp* reporter assays for intron splicing analysis showed that truncation of either F1 or F5 in the *PF16* Intron1 completely diminished the BFP expression in male gametocytes, while loss of either F2, F3, or F4 had no significant effect on the BFP expression in male gametocytes (see the picture in the next page, not included in the revised manuscript).

Considering that the sequences of F1 and F5 are close to the 5' and 3' exon-intron boundary respectively, which are known to be critical for general intron recognition by the spliceosome, deletion of F1 or F5 would much likely impair the intron splicing. The results shown here of investigating the effect of mutagenesis on the splicing of *PF16* Intron1 could not allow for discriminating the effect between the general spliceosome recognition or the specific RBPm1 recognition. Currently, we would not prefer to conclude that F1 and F5 are the regions critical for the RBPm1 recognition.

Together, based on the results from both mutation in the potential RNA motif and mutagenesis in the target intron, we speculate that RBPm1-target introns may possess some sequence-independent features, such as RNA structures or epigenetic modifications, which block the direct access of E complex components but provide RBPm1 binding affinity. In this condition, the spliceosome could only assemble at those introns with the assistance of RBPm1. Further experiments such as CLIP-seq, RNA-SELEX, and functional assays are needed to identify any sequential, structural, or epigenetic signal in the axonemal introns for RBPm1 recognition and binding.

A. Schematics showing the mutagenesis in the RBPM1-target intron *PF16* intron1. Five partly overlapping fragments (F1: 3-71 nt, F2: 50-121 nt, F3: 109-179 nt, F4: 169-230 nt, and F5: 231-274 nt) were deleted and five sequences of *PF16* Intron1 with each fragment truncated (Δ F1, Δ F2, Δ F3, Δ F4, and Δ F5) were obtained.

B. The *bfp* reporter assays for splicing analysis of the truncated *PF16* intron1. A series of transgenic lines were generated with wildtype or truncated *PF16* intron 1 (purple line) - inserted *bfp* cassette integrated at the *p230p* locus of the *DFsc7* line. The results showed that truncation of F1 or F5 in the *PF16* intron1 completely diminished the BFP expression in male gametocytes, while loss of either F2, F3, or F4 had no significant effect on the BFP expression in male gametocytes.

In the model proposed in Figure S11M and the discussion, the authors suggest that the weak transcription of axonemal genes together with IR prevents their protein expression in females. Could there be other explanations for the low transcript levels, like the possibility that there is a similar level of transcription as in males but a faster mRNA decay due to RI?

Response:

Let's suppose that axonemal genes have similar levels of transcripts in female and male gametocytes and that intron retention leads to rapid mRNA degradation. In these situations, we would expect similar levels of axonemal proteins in both female and male gametocytes after removing the retained introns.

However, the results in Fig S11B, D, and F (also shown below) showed that the levels of proteins (PF16, Dlc1, and PY17X_1109100) in female gametocytes were significantly lower than those in male gametocytes after genomic deletion of the retained intron (*PF16* intron1, *dlc1* intron4, and *PY17X_1109100* intron1). Therefore, we speculate that the weak transcription of axonemal genes, coupled with intron retention, may hinder the protein expression in females.

Minor comments:

40: the order should be “escape from host erythrocytes and develop into fertile gametes”. The gametes are not fertile until after egress from the RBC.

Response: Thanks for pointing it out. We changed “the gametocytes develop into fertile gametes and escape from host erythrocytes, a process known as gametogenesis” to “the gametocytes escape from host erythrocytes and develop into fertile gametes, a process known as gametogenesis” in line 40.

81: Substitute "was" with "is." "Was" implies a past state that no longer exists.

Response: we changed “We demonstrated that RBPm1 was a nuclear RBP.....” to “We demonstrated that RBPm1 is a nuclear RBP” in line 83.

107: wild type instead of wildtype

Response: corrected in line 109

110: substitute at for in

Response: corrected in line 112

135: I could not find any data showing that disruption of Rbpml does not affect asexual blood stage proliferation. The data shows that there is no effect on gametocyte formation.

Response:

Using CRISPR/Cas9 we obtained the Rbpml null parasite line ($\Delta Rbpml$) and further quantitated the proliferation of $\Delta Rbpml$ in the asexual blood stage. The results showed normal development of the parasite compared to WT (see the below picture, not included in the revised manuscript).

351-352: It is not clear why the “expected” binding to the “target” intron and not the “neighboring” introns. This could be better clarified for the non-expert reader.

Response: we changed “As expected, RBPm1 bound these target introns but not the neighboring introns or exons” to “As expected, RBPm1 bound these target introns but not

the neighboring introns or exons since each intron is individually excised as a lariat RNA during the splicing”.

404: “significantly more”, could the authors quantify the band intensity and calculate the ratio against the bands in the input lane?

Response: we added the quantitation information of the relative intensity for the bands in the blot in the revised Figure 6O and P and Figure 7H.

458: consider editing to “...entering the mosquito midgut...”

Response: we changed “upon entering into the midgut.....” to “upon entering into the mosquito midgut.....”.

Figure 2D, G, H, I and L: please verify the statistical analysis. T-test and Mann-Whitney are not appropriate. One-way ANOVA with multiple comparison should be used.

Response: we used One-way ANOVA with multiple comparison for the statistical analyses in Figures 2D, G, H, I and L, and also updated the information in the legend in the revised manuscript.

Figure 3B: “The numbers indicate the relative intensities...”, relative intensity to what? BiP? Please clarify.

Response:

The band intensity of Tubulins was calculated as the ratio against the band intensity of BiP in the $\Delta Rbpm1$ group, and the ratio was further normalized with that in the 17XNL group while the ratio was set as 1.0 in the 17XNL group.

Figure 3D: Consider editing the second sentence to “Inset panels show longitudinal sections...”

Response:

In the revised legend of Figure 3D, we changed “Longitudinal sections (d1, d2, d3 at the top panels) and cross sections (d4, d5, d6 at the bottom panels) of axonemes were shown.” to “Inset panels show longitudinal sections (d1, d2, d3 at the top panels) and cross sections (d4, d5, d6 at the bottom panels) of axonemes”.

Reviewer#3

The study “An axonemal intron splicing program sustains Plasmodium male development” by Guan and colleagues describes the function of the RNA-binding protein Rbpm1 during male gametogenesis of the rodent malaria pathogen *Plasmodium yoelii*. The study is very attractive and uses a wide range of state-of-the-art methods, including mutant generation and omics analysis. However, I see several conceptual weaknesses in the study that would call the results into question. These are my comments:

The study is based on the statement that Rbpm1 is expressed in male gametocytes and is important for exflagellation. The authors use immunostaining of alpha and beta tubulin to label male gametocytes. However, the two tubulin proteins are not specific for male gametocytes, but are components of the microtubule cytoskeleton, as it is found in all parasite stages. Microtubules are expressed in female and male gametocytes, including below the plasma membrane. In the study, the authors provide no evidence of male expression of tubulins, nor do they refer to published studies. All subsequent experiments on gender specificity based on this statement must then be questioned.

Response:

α-Tubulin II participates in the formation of axonemal microtubules during male gametogenesis. Since the first report of α-Tubulin II as a specific marker for the *Plasmodium* male gametocytes in 1992, the α-Tubulin II has been widely used as a male gametocyte-specific marker in the *P. falciparum*, *P. berghei* and *P. yoelii*. Below are some of the publications that used the α-Tubulin II as a male gametocyte-specific marker.

PMID	Journal	Year	Title
1484548	Mol Biochem Parasitol	1992	Alpha-tubulin II is a male-specific protein in Plasmodium falciparum
8858179	J Cell Biol	1996	A developmental defect in Plasmodium falciparum male gametogenesis
15935755	Cell	2005	Proteome analysis of separated male and female gametocytes reveals novel sex-specific Plasmodium biology
21173797	Cell Res	2011	Malaria parasites form filamentous cell-to-cell connections during reproduction in the mosquito midgut
27298255	Nucleic Acids Res	2016	Integrated transcriptomic and proteomic analyses of P. falciparum gametocytes: molecular insight into sex-specific processes and translational repression
29042501	mBio	2017	Plasmodium falciparum Calcium-Dependent Protein Kinase 2 Is Critical for Male Gametocyte Exflagellation but Not Essential for Asexual Proliferation
29316140	Cell Microbiol	2018	A male gametocyte osmiophilic body and microgamete surface protein of the rodent malaria parasite Plasmodium yoelii (PyMiGS) plays a critical role in male osmiophilic body formation and exflagellation
30703164	PLoS Pathog	2019	Plasmodium male gametocyte development and transmission are critically regulated by the two putative deadenylases of the CAF1/CCR4/NOT complex

32577509	Sci Adv	2020	Conditional expression of PfAP2-G for controlled massive sexual conversion in Plasmodium falciparum
31634979	Cell Microbiol	2020	Vital role for Plasmodium berghei Kinesin8B in axoneme assembly during male gamete formation and mosquito transmission
32883804	Mol Cell Proteomics	2020	A Comprehensive Gender-related Secretome of Plasmodium berghei Sexual Stages
37704606	Nat Commun	2023	Plasmodium ARK2 and EB1 drive unconventional spindle dynamics, during chromosome segregation in sexual transmission stages
37535401	mBio	2023	Plasmodium microtubule-binding protein EB1 is critical for partitioning of nuclei in male gametogenesis
37288670	J Cell Sci	2023	Radial spoke protein 9 is necessary for axoneme assembly in Plasmodium but not in trypanosomatid parasites

In our experiments in the *P. yoelii*, we used anti- α -Tubulin II antibody to stain the parasites. α -Tubulin-II displays abundant and uniform distribution in the cytoplasm of male gametocytes. In contrast, the α -Tubulin II signal is undetectable or extremely weak in female gametocytes and asexual blood stages (ABS) (see the below picture, not included in the revised manuscript). In addition, we did not use β -Tubulin as the male gametocyte marker in our experiments.

The *Rbp1::6HA* parasites were co-stained with anti- α -Tubulin II antibody (Sigma-Aldrich, Cat#T6199), anti-HA antibody, and Hoechst33342. α -Tubulin-II displays abundant and uniform distribution in cytoplasm of male gametocytes (purple dashed lines). The female gametocytes (orange dashed lines) and asexual blood stages (ABS, white dashed lines) have undetectable or extremely weak α -Tubulin II signals (white arrows). RBPm1 is exclusively localized in the nucleus of male gametocytes with abundant and cytoplasmic α -Tubulin II expression.

To further address the concern of Reviewer#3, we changed “a marker for male gametocytes)” to “a highly expressed protein in male gametocytes” in line 118 in the revised manuscript. In addition, we added a citation (PMID: 1484548. Alpha-tubulin II is a male-specific protein in *Plasmodium falciparum*) which had reported the male-specific expression of alpha-tubulin II in the Plasmodium gametocytes.

During their analyses, the authors silence a number of genes and examine their exflagellation behaviour, such as the genes encoding kinesin 8b and PF16. However, no analyzes of intraerythrocytic growth or gametocytogenesis and female gametogenesis are shown. Of course, if growth rates or gametocytogenesis are already impaired, then this would also have an indirect impact on exflagellation, without the genes being involved in exflagellation at all.

Response:

Using CRISPR/Cas9 we obtained the Rbpm1 null parasite line ($\Delta Rbpm1$) and further quantitated the proliferation of $\Delta Rbpm1$ in asexual blood stage. The results showed normal development of $\Delta Rbpm1$ compared to WT (see the below picture, not included in the revised manuscript).

In our study, we also deleted the genes of *P. yoelii* *Kinesin8B* and *PF16* via CRISPR/Cas9 and easily obtained the clones of mutant parasites, suggesting a non-essential function of *Kinesin8B* and *PF16* in the asexual blood stage. In addition, both the *P. yoelii* $\Delta kinesin8b$ and $\Delta PF16$ produced comparable levels of gametocytes as WT (see the results in Fig S8A). These results are consistent with the phenotypes of the *kinesin8b* and *PF16* mutants previously reported in the *P. berghei* (PMID: 31409625, 31634979, and 20886115).

In the course of the studies on Rbpm1, the authors carried out comparative transcriptome studies on Rbpm1-KO versus wild type. However, the authors rush through the experiment; no information is given on the number of deregulated genes, nor are their GeneIDs and GO terms given. Important genes may be affected that indicate a function of Rbpm1 outside of exflagellation, so the data need to be disclosed.

Response:

Thank the reviewer's suggestion. In the revised manuscript, we added information on the gametocyte transcriptome data (including the *DFsc7* female gametocyte, *DFsc7* male gametocyte, and *DFsc7;ΔRbpm1* male gametocytes). We analyzed the RNA-sequencing quality, number of genes, heat maps for differential expression, and gene ontology.

List of the relevant files added in the zipped source data file:

Source data Figure 1. RNA-seq information and analysis of the transcriptome data

Source data Table 1. List of differentially expressed genes between male and female gametocytes of the *P.yoelii* *DFsc7* line

Source data Table 2. List of differentially expressed genes in male gametocytes between *DFsc7* (Parental) and *DFsc7;ΔRbpm1* (Mutant) lines of the *P.yoelii*

The authors performed BioID analyzes to identify interaction partners of *Rbpm1*. 113 significant proteins are identified, but their names or GeneIDs are not mentioned. Here too, the question arises as to whether there are interaction partners that indicate a function of *Rbpm1* outside of exflagellation. Therefore, the data should also be disclosed here. String analysis would be helpful to visualize protein interaction clusters.

Response:

In the revised manuscript, we added information on the 113 RBPm1-interacting protein candidates from the proteomic data. Furthermore, we analyzed the RBPm1 interacting protein candidates using the String program.

List of the relevant files added in the zipped source data file:

Source data Figure 2. Protein-protein interaction analysis of the RBPm1 interacting protein candidates

Source data Table 3. List of the RBPm1 interacting protein candidates identified in this study

Further comments:

Female gametogenesis is measured and depicted in Figure 2F. However, only one image is shown and it remains unclear how many cells were evaluated.

Response:

In each experiment, more than 30 female gametocytes were analyzed in each group of 17XNL and *ΔRbpm1* parasites. All of the female gametocytes displayed P28 positive after activation. In the Figure 2 legend of the revised manuscript, we added information for the number of cells evaluated.

Co-immunoprecipitations are shown in Figures 6E, F, G. However, it can be seen from the molecular weights and the original blots that the figures do not represent the same blots, but that the bands were put together from different blots. The data is therefore questionable and the negative and loading controls are inconclusive.

Response:

All the blots, including those in Figures 6E, F, G, are strictly the same as those in the raw images of blots. We never manipulated bands from different blots.

To address the concern of the reviewer, the raw blot images with their original films were provided.

In the film, there are two technical replicates of blots (three lanes in each). The band indicated by the asterisk is one band in the technical replicate 1.

The full view of the source data

To save the film usage, we sometimes placed different PVDF membranes on the same film for exposure. In this film, there are placed with two different blots. One blot is for histone H3, the other blot indicated by the asterisk is to detect other protein.

In the film, there are two technical replicates of blots (three lanes in each). The band indicated by the asterisk is one band in the technical replicate 1.

The full view of the source data

In this film, there are placed with several different blots. One blot is for histone H3, and the blot indicated by the asterisk is to detect other protein.

The authors report that all IR genes are male-specific (Fig. S4A). I do not understand, how they confirmed this.

Response:

Fig S4A (also shown below) provided information for the transcript level (normalized counts) of the 26 IR genes in both male and female gametocytes from the transcriptome data. Based on this, we stated “These genes were specifically or preferentially transcribed in the male gametocytes”.

A

26 genes with intron retention							
Molecular function	Gene ID	Total number of introns in gene	# of the retained intron	Length of the retained intron (bp)	Male transcription (normalized counts)	Female transcription (normalized counts)	Ratio (Male/Female)
Basal body & axoneme assembly	kinesin8b (PY17X_0204100)	4	1	239	5148	2	3217
	PF16 (PY17X_0919000)	1	1	276	580	38	15
Axoneme motility	dhc6 (PY17X_0603800)	33	20	241	842	5	160
	dhc7 (PY17X_0510800)	10	7	235	1070	3	318
	dlc1 (PY17X_1241500)	5	4	193	296	68	4
	dlc2 (PY17X_0302800)	3	1	195	503	19	27
	drc1 (PY17X_0721100)	9	2, 3	179, 150	597	93	6
	dbc (PY17X_1333900)	5	1	772	2059	40	51
Function unknown	md2 (PY17X_1450400)	1	1	505	1534	53	29
	PY17X_1109100	3	1	353	1662	55	30
	PY17X_0521800	16	1	278	777	38	20
	PY17X_1311800	13	5	248	1342	3	446
	PY17X_1323900	1	1	322	710	103	7
	PY17X_1357300	7	5, 7	212, 143	2644	4	630
	PY17X_1335600	1	1	338	1320	3	417
	PY17X_1452900	13	1	367	1686	6	295
	PY17X_1122300	2	1	240	161	10	15
	PY17X_0523500	5	2, 3	192, 318	348	1	533
	PY17X_0508900	7	5	380	485	10	49
	PY17X_1320300	3	2	183	195	24	8
	PY17X_0833600	2	2	302	526	12	46
	PY17X_1341200	5	1	208	731	1	803
	PY17X_1305400	15	11, 12	261, 330	1583	14	116
	PY17X_0415900	13	13	171	431	2	201
	PY17X_0105800	1	1	193	107	3	33
PY17X_1216400	10	5	134	712	1	596	

Out of the 26 IR genes, we further chose 12 genes and generated parasite clones with endogenous protein tagged with 6HA. The results verified their specific expression of proteins in male gametocytes (Fig. S6, also shown below).

The abstract and introduction contain many grammatical errors that, surprisingly, are not present in the rest of the text. The introduction is also very general. Generalized statements about the fact that RNA-binding proteins are hardly known in male gametocytes are also not true; some recent studies report such proteins (e.g. Hanhsen et al., 2022; Russell et al., 2023, Farrukh et al., 2023, etc).

Response:

Thank the reviewer for pointing it out. We have a professional check the grammar and language throughout the manuscript.

In the revised abstract, we changed “the role of RBP in defining the *Plasmodium* male transcriptome and its function in the male gametogenesis remain elusive” to “the role of RBP in defining the *Plasmodium* male transcriptome and its function in male gametogenesis remains incompletely understood.”

In the revised introduction, we added some description of RBP with citation and changed “RNA-binding protein (RBP) can interact with mRNA to regulate RNA metabolism and function” to “RNA binding proteins (RBPs) can interact with transcripts in all manner of RNA-driven processes. RBPs regulate all aspects of RNA life, including mRNA transcription, splicing, modification, trafficking, translation, and decay.”

In the section of discussion (paragraphs 2 and 3), we described and cited several RBPs functioning in the gametogenesis of *Plasmodium*, including Hanhsen *et al.*, 2022 and Russell *et al.*, 2023. According to the suggestion of the reviewer, we added the citation of work by Farrukh *et al.* (2023, PMID: 38148574, which was published after the submission of this manuscript).

REVIEWERS' COMMENTS

Reviewer #1 (Remarks to the Author):

The authors have used the publicly available single cell dataset from Russell et al, 2023, to bioinformatically cross-validate the *P. berghei* ortholog of PyRBPm1 (md5). Their analyses indicate md5 regulation of axonemal intron splicing, and that the function of PyRBPm1 is conserved between *P. yoelii* and *P. berghei* due to intron retention in 12 orthologous genes as a result of the RBPm1/md5 knockout. Thus, they have provided information indicating that there is conservation in function between RBPm1 between *P. berghei* and *P. yoelii*. In addition, they used the malaria atlas to show that the *P. falciparum* ortholog (PF3D7_0414500) of PyRBPm1 is largely expressed among male gametocytes, in common with the expression pattern in *Py* and *Pb*. It is highly unfortunate that the authors do not have the capacity to perform functional validations of the RBPm1 ortholog in *P. falciparum*, however it is understandable since they do not have the platform to perform such studies.

In total the authors have gone through some length to computationally cross-validate RBPm1 function between the rodent infectious species *P. yoelii* and *P. berghei*, as well as the RBPm1 gene expression patterns across the three species. Additionally, they have improved the language in the introductory parts of the manuscript. I feel that they have responded to my concerns to the degree that they have been capable.

Reviewer #2 (Remarks to the Author):

The authors have answered all the questions and have edited the manuscript accordingly.

Reviewer #3 (Remarks to the Author):

I am still not d'accord with using alpha-tubulin II as a male marker. While there are several reports stating that alpha-tubulin II is male specific (as listed by the authors), other publications state the contrary (e.g. PMID: 21209927; PMID: 16115694; plus, my own experience with this protein). However, for the publication I would leave it at that.

The authors have included source data, i.e. excel files with lists of genes from the RNA-seq experiments and of proteins from the TurboID analysis, as well as figures with string networks and GO terms. It is unclear to me, though, if these source data are for the reviewers only. In the manuscript, there are no references to these data. Nonetheless, the source data should be included and should be part of the publication, because they include important information and are essential for the reader. Noteworthy, I was not able to open excel tables 1 and 2.

When I checked the interactors of RBPm1/md5 (excel table 3), I noticed that the male/female development proteins as identified by Russell et al., 2023, as well as several interactors of MD3 as identified by Farrukh et al., 2024, are among them. This should be noted in the results and/or discussion section.

The numbers of RNA-seq-based deregulated genes (line 215) should be mentioned.

Response to Reviewer Comments on the manuscript

Reviewer#1

The authors have used the publicly available single cell dataset from Russell et al, 2023, to bioinformatically cross-validate the *P. berghei* ortholog of PyRBPm1 (md5). Their analyses indicate md5 regulation of axonemal intron splicing, and that the function of PyRBPm1 is conserved between *P. yoelii* and *P. berghei* due to intron retention in 12 orthologous genes as a result of the RBPm1/md5 knockout. Thus, they have provided information indicating that there is conservation in function between RBPm1 between *P. berghei* and *P. yoelii*. In addition, they used the malaria atlas to show that the *P. falciparum* ortholog (PF3D7_0414500) of PyRBPm1 is largely expressed among male gametocytes, in common with the expression pattern in *Py* and *Pb*. It is highly unfortunate that the authors do not have the capacity to perform functional validations of the RBPm1 ortholog in *P. falciparum*, however it is understandable since they do not have the platform to perform such studies.

In total the authors have gone through some length to computationally cross-validate RBPm1 function between the rodent infectious species *P. yoelii* and *P. berghei*, as well as the RBPm1 gene expression patterns across the three species. Additionally, they have improved the language in the introductory parts of the manuscript. I feel that they have responded to my concerns to the degree that they have been capable.

Response: We thank the reviewer for the comments.

Reviewer#2

The authors have answered all the questions and have edited the manuscript accordingly.

Response: We thank the reviewer for the comments.

Reviewer#3

I am still not d'accord with using alpha-tubulin II as a male marker. While there are several reports stating that alpha-tubulin II is male specific (as listed by the authors), other publications state the contrary (e.g. PMID: 21209927; PMID: 16115694; plus, my own experience with this protein). However, for the publication I would leave it at that.

The authors have included source data, i.e. excel files with lists of genes from the RNA-seq experiments and of proteins from the TurboID analysis, as well as figures with string networks and GO terms. It is unclear to me, though, if these source data are for the

reviewers only. In the manuscript, there are no references to these data. Nonetheless, the source data should be included and should be part of the publication, because they include important information and are essential for the reader. Noteworthy, I was not able to open excel tables 1 and 2.

Response: Thank the reviewer for the suggestion. This information was included in the publication. In the revised manuscript, we re-arranged the source data tables 1-3 (RNA-seq experiments and the proteins TurboID analysis) as supplementary data 1-3 and added references in the text accordingly.

When I checked the interactors of RBPm1/md5 (Excel Table 3), I noticed that the male/female development proteins as identified by Russell et al., 2023, as well as several interactors of MD3 as identified by Farrukh et al., 2024, are among them. This should be noted in the results and/or discussion section.

Response: In the works by Russell et al., 2023 and Farrukh et al., 2024, they used mass spectrometry and detected GD1-interacting proteins and MD3-interacting proteins. Furthermore, they showed that GD1 and MD3 are localizing in the cytoplasm of gametocytes, which is quite different from the specific nuclear localization of RBPm1/Md5. Currently, we prefer not to speculate or emphasize the interaction between RBPm1/Md5 and GD1-interacting proteins or MD3-interacting proteins in the discussion.

The numbers of RNA-seq-based deregulated genes (line 215) should be mentioned.

Response: Thank the reviewer for the suggestion. We added the information in the legend of the supplementary figure 3H.